# Ezh2 emerges as an epigenetic checkpoint regulator during monocyte differentiation limiting cardiac dysfunction post-MI

Julie Rondeaux[1], Déborah Groussard [1], Sylvanie Renet[1], Virginie Tardif [1], Anaïs Dumesnil[1], Alphonse Chu[2], Léa Di Maria[1], Théo Lemarcis[1], Manon Valet[1], Jean-Paul Henry[1], Zina Badji[3], Claire Vézier[3], Delphine Béziau-Gasnier[3], Annette E. Neele[4], Menno P. J. de Winther [4], Dominique Guerrot[5], Marjorie Brand[2], Vincent Richard [6], Eric Durand[7], Ebba Brakenhielm [1] & Sylvain Fraineau [1] ✉

Epigenetic regulation of histone H3K27 methylation has recently emerged as a key step during alternative immunoregulatory M2-like macrophage polarization; known to impact cardiac repair after Myocardial Infarction (MI). We hypothesized that EZH2, responsible for H3K27 methylation, could act as an epigenetic checkpoint regulator during this process. We demonstrate for the first time an ectopic EZH2, and putative, cytoplasmic inactive localization of the epigenetic enzyme, during monocyte differentiation into M2 macrophages in vitro as well as in immunomodulatory cardiac macrophages in vivo in the post-MI acute inflammatory phase. Moreover, we show that pharmacological EZH2 inhibition, with GSK-343, resolves H3K27 methylation of bivalent gene promoters, thus enhancing their expression to promote human monocyte repair functions. In line with this protective effect, GSK-343 treatment accelerated cardiac inflammatory resolution preventing infarct expansion and subsequent cardiac dysfunction in female mice post-MI in vivo. In conclusion, our study reveals that pharmacological epigenetic modulation of cardiac-infiltrating immune cells may hold promise to limit adverse cardiac remodeling after MI.

Myocardial infarction (MI) triggers an endogenous wound healing response including an early innate immune response involving myeloid cells infiltrating cardiac tissue. Macrophages originating from marrow-derived circulating monocytes, are key regulators of cardiac repair post-MI[1]. Indeed, rapidly after MI, circulating classical, pro-inflammatory Ly6C[hi] monocytes are recruited into the heart and accumulating in the ischemic area where they differentiate into pro-inflammatory M1-like macrophages that contribute to the early inflammatory phase within the first 3 days post-MI in mice. Over the course of cardiac repair, during the first week after MI, the inflammatory phase gives way to the repair phase, characterized by a switch of pro-inflammatory macrophages toward an alternative,

[1]Univ Rouen Normandie, Inserm EnVI UMR 1096, F-76000 Rouen, France. [2]Sprott Center for Stem Cell Research, Regenerative Medicine Program, Ottawa Hospital Research Institute, General Hospital, Mailbox 511, 501 Smyth Road, Ottawa ON K1H8L6, Canada. [3]CHU Rouen, Department of Cardiology, F-76000 Rouen, France. [4]Department of Medical Biochemistry, Amsterdam Cardiovascular Sciences, Amsterdam Institute for Infection and Immunity, Amsterdam UMC, University of Amsterdam, Amsterdam, Netherlands. [5]Univ Rouen Normandie, Inserm EnVI UMR 1096, CHU Rouen, Department of Nephrology, F-76000 Rouen, France. [6]Univ Rouen Normandie, Inserm EnVI UMR 1096, CHU Rouen, Department of Pharmacology, F-76000 Rouen, France. [7]Univ Rouen Normandie, Inserm EnVI UMR 1096, CHU Rouen, Department of Cardiology, F-76000 Rouen, France. ✉e-mail: sylvain.fraineau@univ-rouen.fr

immunomodulatory Ly6C[lo] M2-like macrophage phenotype[2]. Although cardiac-recruited Ly6C[hi] monocytes initially differentiate towards pro-inflammatory macrophages, they gradually change their differentiation program towards reparative immunomodulatory macrophage subpopulations[2]. Importantly, this macrophage phenotype transition promotes angiogenesis, lymphangiogenesis and extracellular matrix synthesis and deposition necessary to create a mature fibrotic scar. The proper sequence of inflammatory and reparative phases is critical to prevent cardiac remodeling and dysfunction to limit development of heart failure. Experimental studies have shown that defective post-MI cardiac healing is induced either by reduced cardiac monocyte infiltration[3], absence of a proper pro-inflammatory phase[4], or sustained duration of the acute inflammatory phase due to pro-inflammatory macrophage accumulation[5]. Similar results were obtained with reduced monocyte to immunomodulatory macrophage differentiation during the reparative phase[6,7]. Indeed, rapid resolution of the inflammatory phase, giving way to an early and strong reparative phase, correlates with reduction of infarct scar expansion and better preservation of cardiac function.

In this context, macrophage phenotype switching, from pro-inflammatory to immunomodulatory, plays a key role in cardiac inflammation and repair post-MI. Accumulating evidence points to a specific epigenetic histone modification, histone H3 lysine 27 tri-methylation (H3K27me3), as an important mechanism to regulate macrophage activation and polarization[8–10]. The repressive H3K27me3 mark is generated by the Polycomb repressive complex 2 (PRC2), containing the epigenetic histone methyltransferase enzyme Enhancer of zeste homolog 2 (Ezh2). Conversely, H3K27 methylation is actively antagonized by two histone demethylases: Jumonji domain-containing protein D3 (Jmjd3) and Ubiquitously transcribed TPR on X (Utx). This fine-tuned balance between activators and repressors of epigenetic modifications on H3K27 impacts the expression of many genes involved in diverse biological processes including cell-fate commitment during cellular differentiation. Jmjd3 has been described as an essential regulator of M2-like immunomodulatory macrophage polarization through upregulation of *Arg1, Chi3l3, Retnla,* and *CCL17* linked to *Irf4* upregulation during helminth infection[11] as well as following IL4 stimulation in both human monocytes[12] and murine macrophages[13]. Although Jmjd3 implication during M2 macrophage polarization is well established, the role of its antagonistic partner, Ezh2, has not yet been well established in macrophages. In contrast, Ezh2 has been suggested to play important roles in regulating inflammatory cell functions in B cells[14,15], T cells[16], neutrophils, and dendritic cells[17]. However, recent reports suggest that Ezh2 may influence macrophage activation by suppressing the expression of anti-inflammatory genes[18] and reducing pro-inflammatory cytokines secretion, leading to a reduction of macrophage-dependent disease development[19,20].

Altogether, these findings suggest that H3K27 methylation promotes a pro-inflammatory state, while H3K27 demethylation promotes an immunomodulatory phenotype in macrophages.

We hypothesized that Ezh2, by modulating H3K27 methylation, may directly regulate monocyte to M2-like immunomodulatory macrophage differentiation and thus represents an attractive therapeutic target to design new pharmacological epigenetic inhibitors to promote cardiac repair post-MI and prevent the occurrence of heart failure.

Here we demonstrate that Ezh2 serves as an epigenetic checkpoint regulator during monocyte to macrophage differentiation. Specifically, translocating EZH2 from the nucleus to the cytoplasm, releases the repression of bivalent genes essential for M2-like immunomodulatory macrophage differentiation and function. Moreover, we promisingly demonstrate that pharmacological Ezh2 inhibition, with GSK-343, reduced the H3K27me3 mark specifically at the promotor of bivalent genes (such as *DLL1, VEGFA, IRF4*) resulting in accelerated resolution of the inflammatory phase, reduced infarct scar expansion, and improved cardiac function post-MI in vivo.

## Results

### Cytoplasmic translocation of Ezh2 in cardiac M2-like immunomodulatory macrophages post-MI

We first investigated the subcellular localization of Ezh2 in cardiac-infiltrating myeloid cells in mice during post-MI cardiac repair. As expected, we observed a nuclear localization of Ezh2 in both CD11b[+]/CD68[-] and Cd11b[+]/Cd68[+] cardiac monocytes (Fig. 1a), and cardiac pro-inflammatory macrophages including in Cd68[+]/iNos[+] (Fig. 1b), Cd68[+]/Cd86[+] (Fig. 1c and Fig. S1), Cd68[+]/MHCII[+] (Fig. 1d and Fig. S2) and Cd68[+]/Il1b[+] (Fig. S3). Unexpectedly, Ezh2 had translocated to the cytoplasm in immunomodulatory macrophages including in Cd68[+]/Cd206[+] (Fig. 1e, f and Fig. S4), Cd68[+]/Trem2[+] (Fig. S5) Cd68[+]/Lyve1[+] (Fig. 1g and Fig. S6) in both sham and post-MI hearts. These observations suggest that cytoplasmic translocation of Ezh2 might induce macrophage differentiation by initiating a switch from pro-inflammatory toward immunoregulatory phenotypes, which may impact the transition from the pro-inflammatory towards the reparative phase post-MI.

### Ezh2 cytoplasmic translocation promotes M2 macrophage polarization in vitro

To better understand the role of Ezh2 cytoplasmic translocation observed in cardiac macrophages post-MI, we reproduced myeloid cell differentiation in vitro starting from peripheral blood circulating monocytes to differentiated M1 or M2 polarized, mature macrophages. Peripheral blood circulating monocytes were subjected to negative magnetic selection before M-CSF-induced in vitro differentiation into non-polarized M0 macrophages, followed by either LPS-induced M1 macrophage polarization, or Il4 and Il10-induced M2 macrophage polarization. We confirmed our previous in vivo observations of a nuclear localization of Ezh2 in both Ly6c[hi] and Ly6c[lo] non-adherent (Fig. 2a and Fig. S7a) and adherent (Fig. 2b and Fig. S7b) monocytes, and in non-polarized M0 (Fig. 2c) and M1 (Fig. 2d) polarized macrophages. In contrast, in agreement with in vivo observation, Ezh2 was found in the cytoplasm in M2 macrophages (Fig. 2e). Interestingly, Ezh2 sub-cellular localization was similar in all monocytes, independent of their polarization status of either classical or non-classical monocyte sub-population phenotype, as assessed by Ly6c expression (Fig. 2a and Fig. S7a). Similarly, the cytoplasmic Ezh2 localization was also found in selective M2a or M2c macrophage differentiation following single cytokine-induced polarization (Fig. S8). Thus, this mechanism seems to be global rather than restricted to specific monocyte or M2 macrophage sub-populations. We hypothesized that Ezh2 outsourcing from the nucleus may be critical to selectively induce M2 macrophage polarization, and that it acts as an epigenetic checkpoint to prevent differentiation and polarization into immunoregulatory alternative M2-like macrophages.

To test this hypothesis, we induced macrophage differentiation in vitro, and examined Ezh2 cytoplasmic translocation as well as Cd206 expression kinetics. We observed an early Ezh2 nuclear to cytoplasmic translocation occurring within 3 h after initiation of M2 macrophage polarization (Fig. 2f) prior to the Cd206[−] to Cd206[+] phenotype/identity switch, which appeared more than 6 h after induction of polarization. We therefore conclude that Ezh2 might actively repress the M2 macrophage transcriptional programming during myeloid cell differentiation, which is relieved by nuclear export of Ezh2 to the cytoplasm.

### M2 macrophage differentiation is epigenetically repressed by Ezh2 control of bivalent gene expression

Ezh2 is known to act as a transcriptional repressor of bivalent genes, characterized by the concomitant presence of both repressive Ezh2-dependent H3K27me3 and activating H3K4me3 epigenetic marks at

their promoter regions[21,22]. We decided to identify which bivalent genes may be possible direct targets of Ezh2 in macrophages. First, we exploited previously published ChIP-sequencing (ChIP-seq) data from CD14+ sorted human monocytes to establish a list of potential EZH2-regulated bivalent genes in human monocytes (Supplementary Data 1)[23,24]. Next, these putative EZH-2 regulated bivalent genes were validated in CD14+sorted human monocytes by assessing the presence of the H3K27me3 repressive mark at the promoter of inactive genes, such as *PAX7* (Fig. 3a, top panel), or the H3K4me3 active mark at the promoter of active genes, such as *ELP3* (Fig. 3a, second panel). Finally, we established the full list of bivalent genes in CD14+ sorted human monocytes (Supplementary Data 1), which included *DLL1*, *VEGFA,* and *IRF4* (Fig. 3a). Gene Ontology (GO) analysis indicated that our list of

bivalent genes in CD14+ sorted human monocytes are enriched in categories related to cardiac remodeling (e.g., cardiac right ventricle morphogenesis, heart development, ventricular septum morphogenesis), vascular remodeling (e.g., blood vessel development, angiogenesis), but also cell migration, wound healing, and cell fate commitment (Fig. 3a). This suggests that de-repression, through EZH2 inactivation, of bivalent genes may promote cardiac, vascular and lymphatic repair functions in monocytes and macrophages post-MI. To investigate, we further examined the baseline bivalent status of selected genes from our ChIP-qPCR analysis in our model of circulating monocytes isolated from human peripheral blood. As expected, we observed a significant enrichment of the repressive H3K27me3 mark in the promoter region of *DLL1*, *VEGFA*, and *IRF4* bivalent genes, as well as in the promoter of

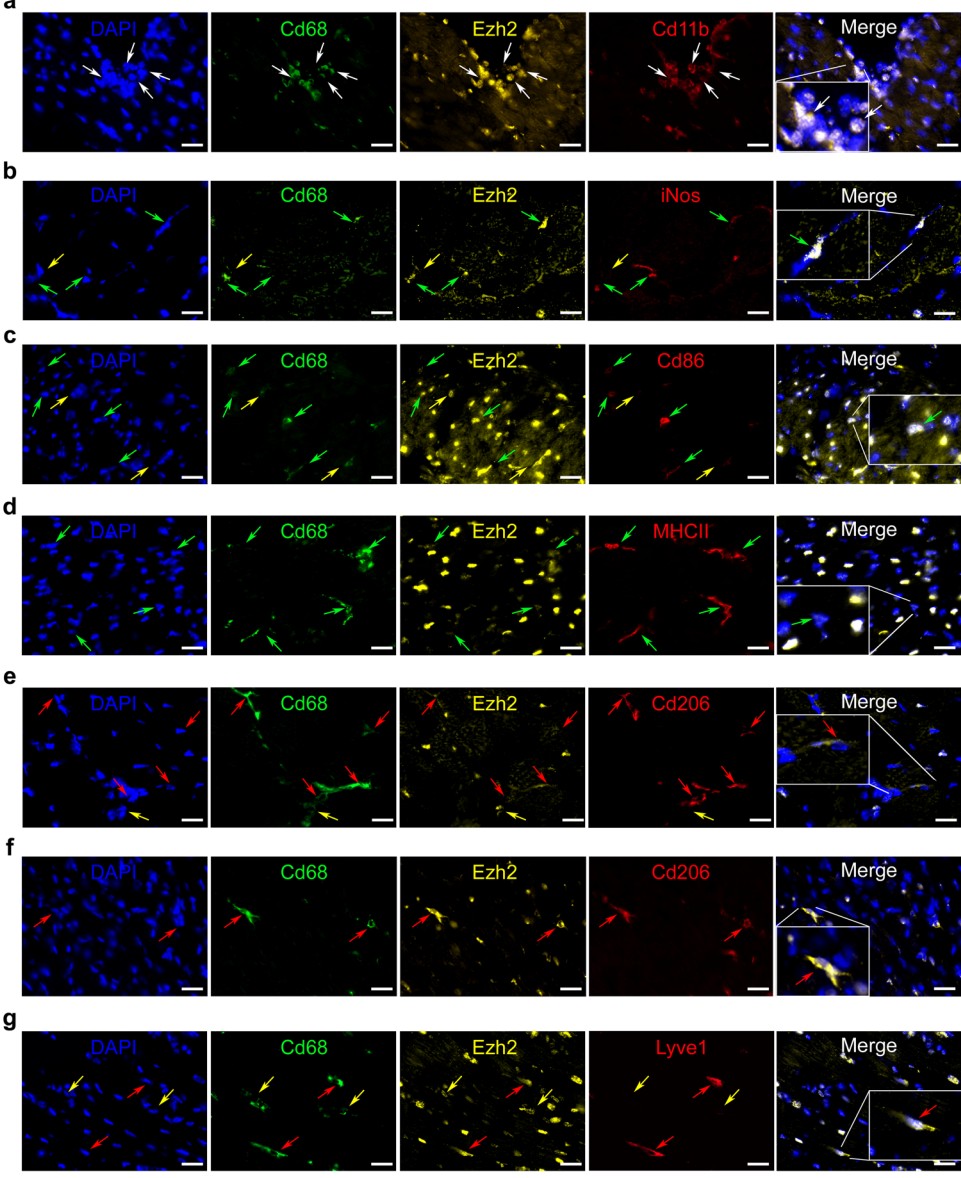

**Fig. 1 | Ezh2 is translocated to the cytoplasm in cardiac M2-like immunomodulatory macrophages in vivo.** Representative pictures of cardiac immunostaining for Ezh2 in **a** Cd11b⁺/Cd68⁻ monocytes, **b** Cd68⁺/iNos⁺, **c** Cd68⁺/Cd86⁺, **d** Cd68⁺/MHCII⁺ pro-inflammatory differentiated macrophages, **e** Cd68⁺/Cd206⁺ immunomodulatory differentiated macrophages at 24 h after coronary ligation in mice to induce MI or **f** Cd68⁺/Cd206⁺, **g** Cd68⁺/Lyve1⁺ cardiac-resident immunomodulatory in healthy sham mice. Nuclei were stained with DAPI (blue), myeloid markers were Cd11b (red, panel **a**), Cd68 (green, panels **a**–**g**), iNos (red, panel **b**), Cd86 (red, panel

**c**), MHCII (red, panel **d**) Cd206 (red, panels **e** and **f**) and Lyve1 (red, panel **g**). Ezh2 (yellow) cellular localization was observed in each cell type but only appeared in immunomodulatory macrophages cytoplasm. Arrows indicate monocytes (white), non-determined macrophages (yellow), pro-inflammatory (green) and immunomodulatory macrophages (red), scale bars represent 25 µm. These cardiac immunostainings have been reproduced at least on three different mice with similar results.

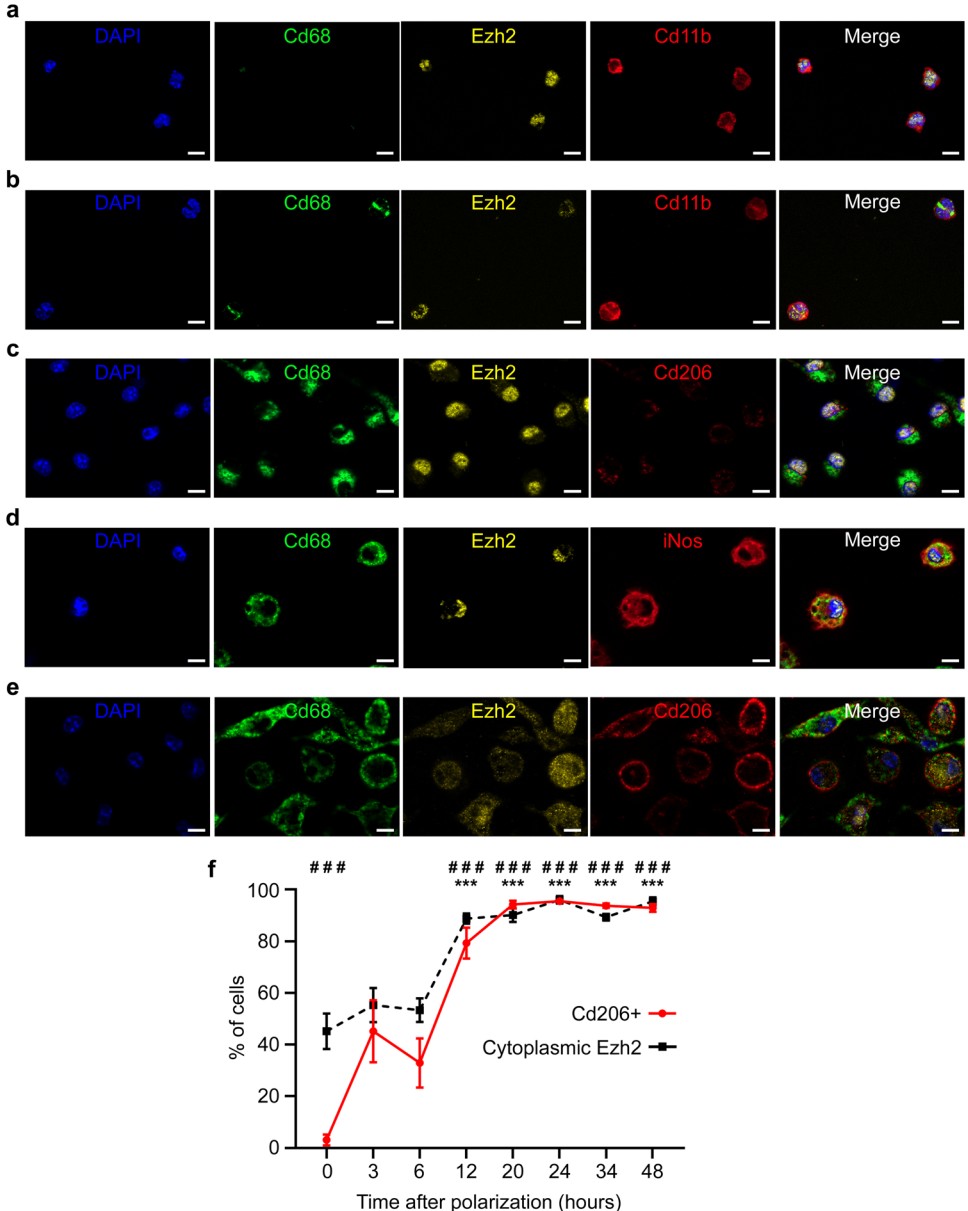

**Fig. 2 | Ezh2 is translocated to the cytoplasm in myeloid cells during M2 polarization in vitro.** Representative pictures of immunostaining for Ezh2 in **a** non-adherent monocytes, **b** adherent monocytes, **c** non-polarized M0 macrophages, **d** M1 macrophages and **e** M2 macrophages, differentiated and polarized in vitro. Nuclei were stained with DAPI (blue), monocytes were identified based on Cd11b (red, panels **a** and **b**) expression, and macrophages based on Cd68 (green, panels **a**–**e**) expression. M1 or M2 macrophage phenotype was determined by expression of iNos (red, panel **d**) or Cd206 (red, panels **c** and **e**). Ezh2 (yellow) cellular localization was observed in each cell type. Scale bars represent 10 μm. These immunostainings have been reproduced at least 3 times from different mice with similar results. Kinetics of Ezh2 subcellular localization and Cd206 expression (**f**) were assessed by immunostaining during M2 macrophage polarization. Data are represented as percentage of total cells mean values ± SEM of 3 independent experiments ($n = 3$) each performed in duplicate for all time points. Asterisk (*) symbol indicates statistically significant difference between cells with either nuclear or cytoplasmic Ezh2 localization. Hashtag (#) depicts significant difference between Cd206 negative and positive cells. ### and ***$p < 0.001$; Two-way ANOVA with Sidak's multiple comparisons test. Source data are provided as a Source Data file.

*PAX7* inactive gene (Fig. 3c). Conversely, we found a significant enrichment of the activating H3K4me3 mark in the promoter of *DLL1*, *VEGFA*, and *IRF4* bivalent genes as well as in the promoter of *ELP3* active gene. These data corroborate the bivalent gene status of *DLL1*, *VEGFA*, and *IRF4* in circulating human monocytes.

## Pharmacological inhibition of EZH2 enhances bivalent gene expression to influence monocyte function in vitro

To determine the functional role of EZH2 on bivalent gene expression in monocytes, we applied epigenetic pharmacological inhibition of EZH2 methyltransferase activity based on GSK-343 treatment in vitro. Once optimal inhibitory conditions (Fig. S9a) and absence of nucleocytoplasmic translocation of EZH2 (Fig. S9b) with GSK-343 were determined, we investigated by RNA-sequencing (RNA-seq) the transcriptomic impact of GSK-343 in isolated human monocytes. This experiment, performed using independent pools of selected peripheral blood monocytes from three distinct non-coronary disease patients (Supplementary Table 1), identified 85 statistically upregulated genes and 48 statistically downregulated genes as compared to vehicle control (Fig. 4a). GO analysis indicated that the 48

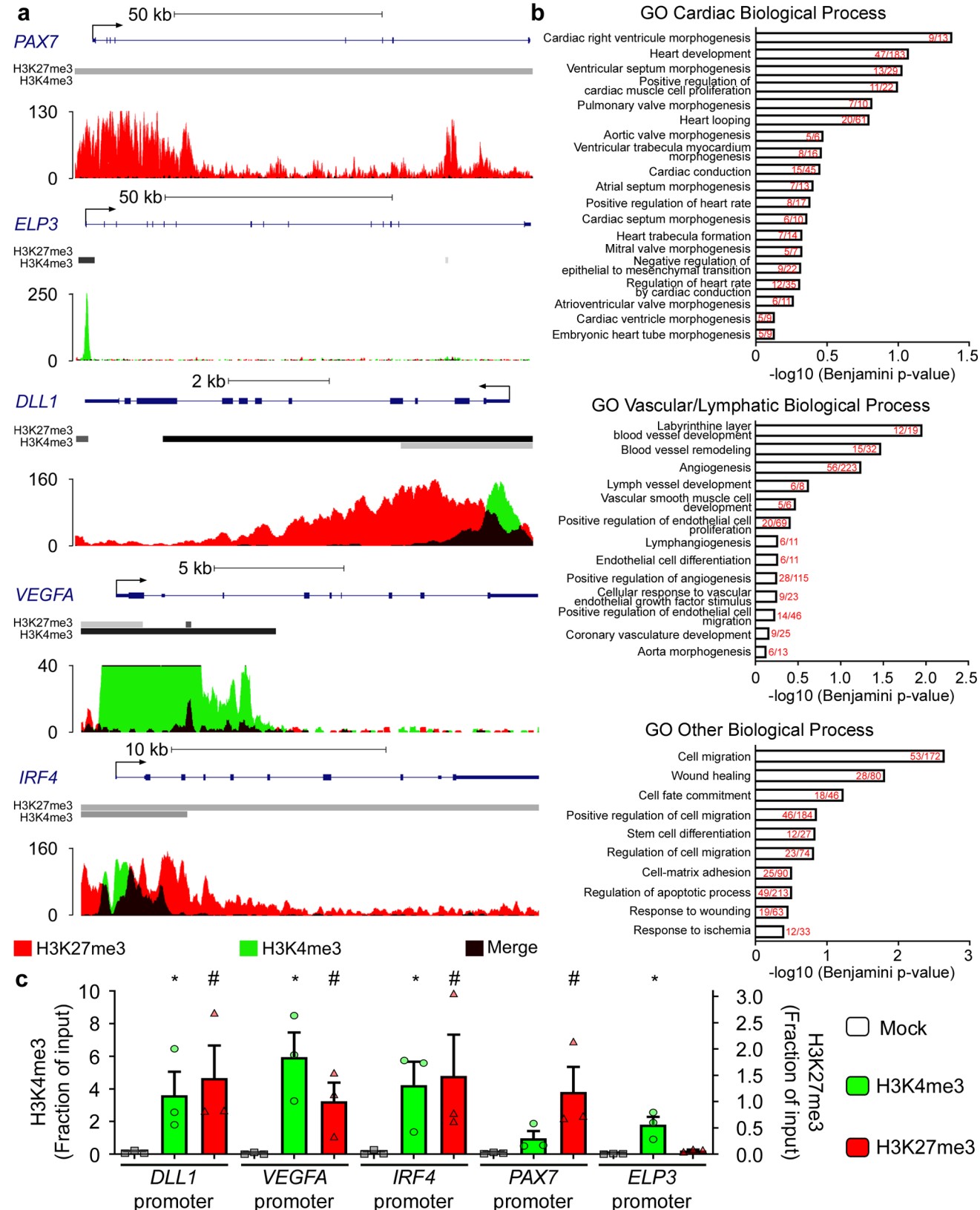

downregulated genes are enriched in categories related to collagen binding, heart development, interleukin-1 binding, wound healing, and negative regulation of transcription (Fig. S10a and Supplementary Data 2). In contrast, upregulated genes are enriched in categories related to chemotaxis, response to hypoxia, angiogenesis, and inflammatory response (Fig. S10b and Supplementary Data 3). The full

lists of down- and upregulated genes and GO categories are available respectively in Supplementary Data 2 and 3.

To get more insight into the molecular mechanisms and identify direct target genes of EZH2 modulation by GSK-343, we compared the list of bivalent genes identified in CD14+ human monocytes, established by ChIP-seq analysis, and the list of GSK-343-induced

**Fig. 3 | Ezh2-regulated bivalent genes in monocytes are implicated in cardio-vascular repair processes.** Representative tracks and peak calling for promotors of inactive (*PAX7*), active (*ELP3*) and bivalent (*DLL1, VEGFA, IRF4*) genes (**a**) generated by ChIP-Seq analysis of human CD14+ monocytes. Representative Gene Ontology (GO) Biological Process categories significantly enriched for bivalent genes in human CD14+ monocytes. The bar graph represents the −log10 (Benjamini *p*-value), obtained from DAVID gene-enrichment in functional annotation terms after Fisher's Exact test filled with the number of genes revealed by the analysis within the number of overall genes in each GO biological process category

indicated in red (**b**). Quantification of H3K27me3 inactive (red) versus H3K4me3 active (green) histone marks in the promoter regions of three bivalent and two non-bivalent genes, assessed by ChIP-qPCR (**c**). Data represent mean fractions of input ± SEM of three independent experiments, corresponding to three different human donors, performed in duplicate (*n* = 3). Asterisk (*) symbol indicates H3K4me3 statistically significant enrichment compared to the inactive *PAX7* promoter (Kruskal–Wallis test). Hashtag (#) depicts H3K27me3 significant enrichment compared to the active *ELP3*: *p < 0.05 and # p < 0.05. Source data are provided as a Source Data file.

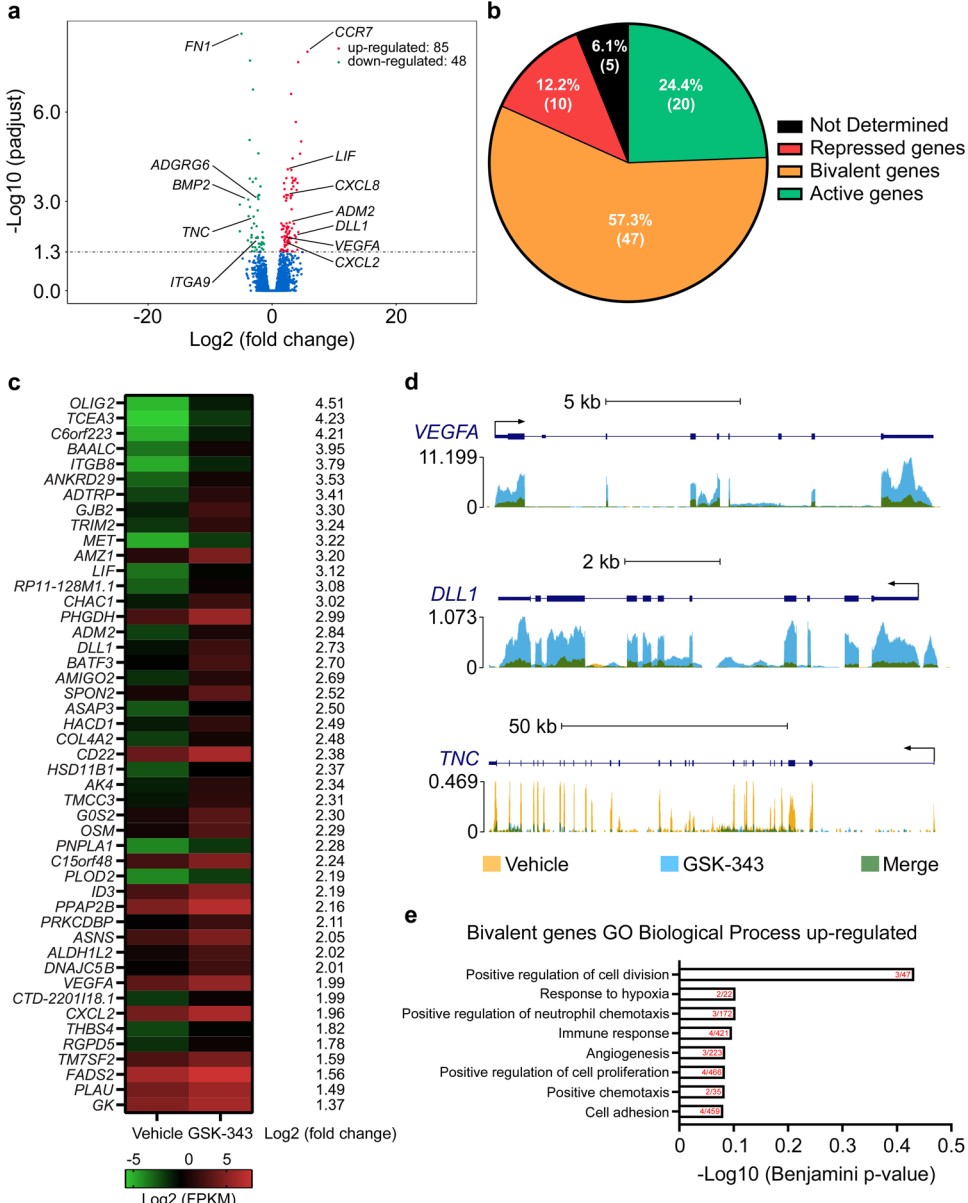

**Fig. 4 | Ezh2 pharmacological inhibition with GSK-343 increases bivalent gene expression and enhances human monocyte homing and angiogenic functions in vitro.** Global changes in gene expression upon GSK-343 treatment versus vehicle treatment analyzed by mRNA-seq (**a**) in cultured human monocytes. Volcano plot shows upregulated (red) and downregulated (green) genes (combined data from three different donors, *n* = 3) considered as statistically significant with a Benjamini-Hochberg adjusted *p*-value < 0.05. Classification of GSK-343-induced genes according to promoter status (**b**). Heatmaps of all significantly upregulated bivalent genes identified by RNA-seq analysis in human monocytes after GSK-343 treatment (**c**). Data obtained from three independent donors (*n* = 3) are expressed in log2

(FPKM). Representative tracks of changes induced by GSK-343 treatment on select up- and downregulated genes in human monocytes (**d**). Representative Gene Ontology (GO) Biological Process categories significantly (Benjamini-Hochberg adjusted *p*-value < 0.05) enriched for GSK-343-induced bivalent genes in human monocytes. The bar graph represents the −log10 (Benjamini-Hochberg adjusted *p*-value), obtained from DAVID gene-enrichment in functional annotation terms after Fisher's Exact test filled with the number of genes revealed by the analysis within the number of overall genes in each GO biological process category indicated in red (**e**).

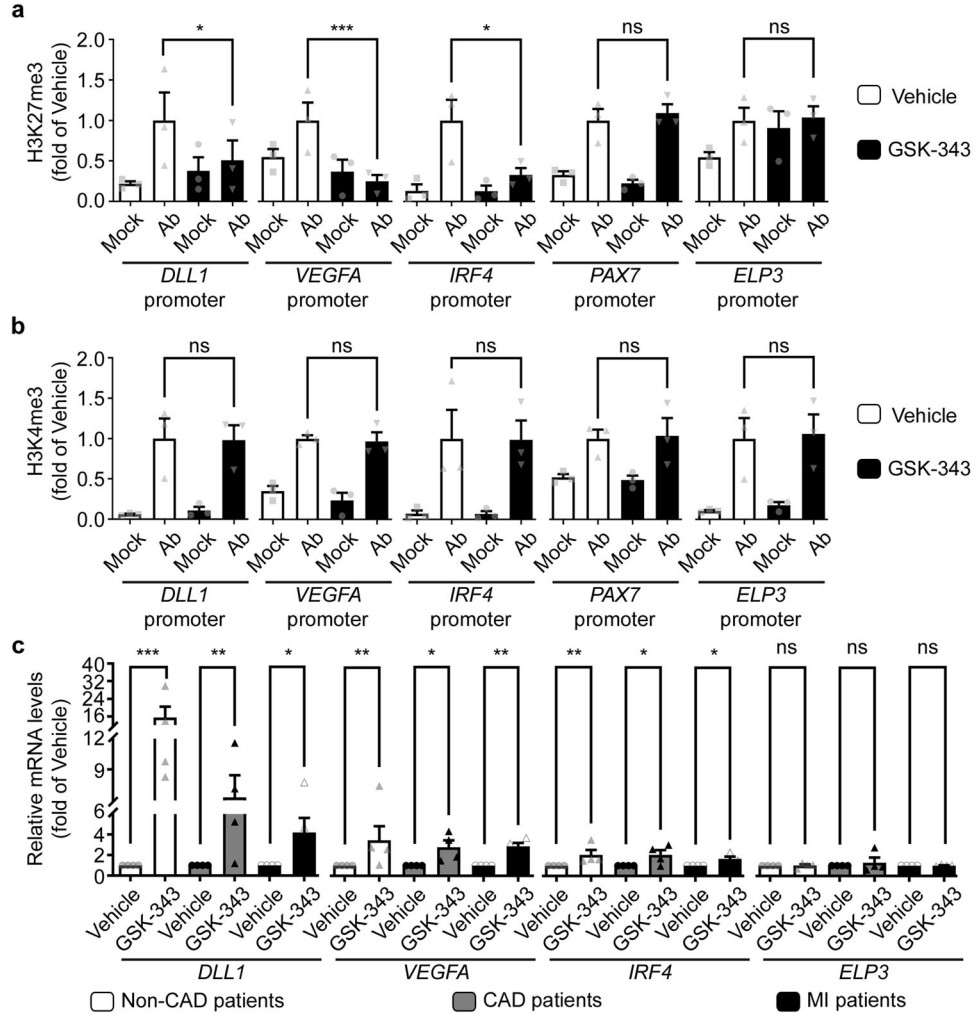

**Fig. 5 | Pharmacological inhibition of EZH2 increases specifically bivalent gene expression through H3K27me3 demethylation in monocytes in vitro.** Enrichment of selected bivalent (*DLL1, VEGFA, IRF4*), inactive (*PAX7*) and active (*ELP3*) gene promoters in H3K27me3 (**a**) and H3K4me3 (**b**) as assessed by ChIP-qPCR. Ab indicates the enrichment obtain with either H3K27me3 (**a**) or H3K4me3 (**b**) targeting antibodies, while Mock represents the data obtain after immunoprecipitation with control IgG. Data are represented as mean fractions of input normalized to vehicle-treated monocytes ± SEM of three independent experiments corresponding to three different donors performed in duplicate (*n* = 3). The *p* values determined by Kruskal–Wallis test are depicted as asterisks in the figures as follows:

\*\*\**p* < 0.001; \*\**p* < 0.01; \**p* < 0.05; ns: non-significant enrichment compared to vehicle-treated monocytes. Transcript levels of indicated genes were measured by RT-qPCR following treatment with vehicle or GSK-343 of selected monocytes from non-coronary patients, CAD or AMI patients (**c**). RT-qPCR values are expressed as mean percentages of vehicle-treated monocytes ± SEM with *B2M* serving as internal control of four independent experiments corresponding to four different donors per group performed in duplicate (*n* = 4). The *p* values are depicted as asterisks after Kruskal–Wallis test in the figure as follows: \*\*\**p* < 0.001; \*\**p* < 0.01; \**p* < 0.05; ns: non-significant. Source data are provided as a Source Data file.

upregulated genes, identified by RNA-Seq, in circulating human monocytes. We found that the majority of genes upregulated following GSK-343 treatment were indeed bivalent genes (47 out of 85 altered genes; 57.3%). In contrast, the proportion of either active (20 out of 85 genes; 24.4%) or repressed (10 out of 85 genes; 12.2%) genes, based on the unique presence of H3K4me3 or H3K27me3 at their promoter level respectively, was minor, with some genes (5 out of 85 genes; 6.1%) for which epigenetic status remains uncharacterized (Fig. 4b). Interestingly, within the established full list of EZH2 direct target genes in monocytes (Fig. 4c), we identified *VEGFA* (>5-fold upregulated by GSK-343), a well-known activator of angiogenesis and vasculogenesis, and *DLL1* (>6-fold upregulated by GSK-343), a member of NOTCH signaling pathway regulating vascular morphogenesis and remodeling (Fig. 4d). Among the 48 downregulated genes, we noted *TNC* (~8-fold reduced by GSK-343), involved in cardiac fibrosis and acceleration of adverse ventricular remodeling post-MI (Fig. 4d). Altogether our RNA-seq data identified a pool of 47 EZH2 bivalent gene targets (Fig. 4b, c)

implicated in chemotaxis (*VEGFA, MET, THBS4*, and *CXCL2*), response to hypoxia (*PLOD2, VEGFA*, and *PLAU*), immune response (*LIF, CD22, OSM*, and *CXCL2*) and angiogenesis (*VEGFA, ADM2, DLL1*, and *COL4A2*) (Fig. 4e) suggesting that EZH2 inhibition with GSK-343 may promote cardiac repair function by myeloid cells.

### EZH2 pharmacological inhibition specifically resolves H3K27me3 level at bivalent gene promoters to activate transcription

To verify that the upregulated expression of bivalent genes in response to GSK-343 treatment was directly due to alleviation of EZH2 gene repression, we next investigated the level of the repressive H3K27me3 chromatin mark in the promotor regions of select target genes. Indeed, we found, using ChIP-qPCR analysis, that in vitro treatment of human monocytes for 72 h with GSK-343 decreased H3K27me3 levels at *DLL1, VEGFA,* and *IRF4* gene promoters (Fig. 5a). Interestingly, H3k27me3 levels were not modified at inactive gene promoters, such

as *PAX7*, nor in constitutive active gene promoters, such as *ELP3* (Fig. 5a). As expected, the levels of the H3K4me3 active mark were not modified upon GSK-343 treatment in the promotor regions of either bivalent genes (e.g. *DLL1*, *VEGFA,* and *IRF4)* or active genes (*ELP3*). This epigenetic activating mark was also not detectable at inactive gene promoters (*PAX7*), in line with unaltered suppressed transcription activity in monocytes (Fig. 5b). Taken together, these results demonstrate the specificity of pharmacological inhibition of EZH2 as an approach to remove epigenetic repressive marks selectively at bivalent gene promoters, without affecting their activating epigenetic marks.

To gain further insight into the potential therapeutic clinical applications of EZH2 inhibition post-MI, we isolated circulating monocytes from patients divided into three groups corresponding to (1) patients without coronary disease; (2) patients diagnosed with stable coronary artery disease (CAD); and (3) patients admitted after acute MI (AMI). Patient characteristics are detailed in Supplementary Table 2. In line with previous findings, we observed a slight albeit non-significant increase, in circulating monocytes levels post-MI, as compared to non-coronary patients (Fig. S11). Monocytes collected from the three patient groups were then treated with GSK-343 in vitro, as described above, followed by RT-qPCR analysis of gene expression of select bivalent (*DLL1*, *VEGFA*, *IRF4*) or active (*ELP3*) genes to assess epigenetic effects. Interestingly, we observed a significant upregulation by GSK-343 of all selected bivalent genes in all three patient groups, while the expression of constitutively active genes remained unaltered (Fig. 5c). These results demonstrate that GSK-343-mediated EZH2 inhibition is able to selectively activate bivalent gene expression in human monocytes irrespective of the underlying disease conditions. Other GSK-343 target genes, identified in our RNA-seq study were confirmed by RT-qPCR, including *ADM2*, *BMP2*, *CXCL2*, *FN1*, and *TNC* (Fig. S12). Moreover, a targeted transcriptomic array of a murine monocyte cell line, focused on assessment of genes selectively expressed in different myeloid cell lineages, revealed that treatment with GSK-343 induces an expression profile similar to M2 macrophages (Fig. S13a). These M2-like promoting effects of GSK-343 were confirmed in both mouse (Fig. S13b, c) and human (Fig. S13d, e) freshly isolated monocytes, as compared with in vitro differentiated M0, M1 or M2 macrophage subtypes. Altogether our in vitro data show that EZH2 acts as an epigenetic check point inhibiting, at the chromatin level, bivalent gene expression to prevent monocyte/macrophage differentiation into an M2-like immunomodulatory cell type. Conversely, treatment of monocytes with a pharmacological inhibitor of EZH2 activity suppressed the H3K27me3 repressive epigenetic mark selectively at the promoter of bivalent genes, thus enhancing their transcription and initiating a cell differentiation program and/or a cell fate commitment into M2-like immunoregulatory macrophages. These observations argue for a potential use of GSK-343 as an epigenetic treatment to accelerate inflammatory resolution after MI to limit cardiac dysfunction and heart failure development.

## GSK-343 accelerates inflammatory resolution and prevents aggravation of cardiac dysfunction in a mouse model of MI

Next, we evaluated in vivo the effects of GSK-343 treatment on cardiac inflammation and cardiac function in a mouse model of MI induced by permanent left coronary ligation. As GSK-343 is a poorly soluble epigenetic drug, we used cyclodextrin-based Captisol®, as previously described for daily intraperitoneal injections[25]. Because our in vitro data indicated that EZH2 inhibition promotes immunomodulatory functions also in monocytes, we first examined circulating and cardiac immune cell levels post-MI using flow cytometry. In line with previous reports, circulating as well as cardiac classical pro-inflammatory monocyte populations were found to be increased 3 days post-MI. Interestingly, GSK-343 treatment significantly reduced classical blood monocyte (Ly6C^hiCx3cr1^hi)/inflammatory

(M1)-like levels, while increasing cardiac levels of the same population in both infarct scar and border zone at 3 days post-MI (Fig. 6a, left and middle panels and Fig. S14). This indicates that Ezh2 inhibition may accelerate cardiac inflammatory kinetics after MI. In agreement, we found that at 8 days post-MI the non-classical monocyte (Ly6C^loCx3cr1^hi)/M2-like immunomodulatory population was significantly increased in both infarct and Border Zone (BZ) areas in GSK-343-treated as compared to vehicle-treated mice (Fig. 6a, right panel and Fig. S14). Notably, the levels of these alternative monocytes in GSK-343-treated mice were similar as in sham controls, arguing for accelerated resolution of inflammation with GSK-343. In contrast, total cardiac macrophages were not significantly increased in GSK-343-treated mice as compared to vehicle controls (Supplementary Table 3 and Fig. S15). To further characterize the cardiac inflammatory response, we assessed cardiac pro-inflammatory cytokine expression by RT-qPCR. In support of faster resolution of inflammation upon GSK-343 treatment, we observed significantly reduced expression of *Il1b* and *Il6* at 8 days post-MI in treated mice, as compared to vehicle-treated MI mice (Fig. 6b). Moreover, MI-induced increase cardiac *Ccl2* (a.k.a monocyte chemoattractant protein MCP1) expression, was also significantly decreased in GSK343-treated mice, while *Ccl21* expression was increased. This may lead to reduced monocyte recruitment and/or increased immune cell efflux via lymphatics in GSK-343 treated mice in the later phase post-MI (Fig. 6b). Interestingly, we did not observe any modification of these genes at 3 days post-MI (Supplementary Table 4).

Altogether, our data argue for accelerated resolution of the inflammatory phase post-MI with GSK-343 treatment. Further, as expected from our RNA-seq data in human monocytes, cardiac bivalent gene expression including *Dll1*, was restored with epigenetic drug treatment while the expression of both *Fn1* and *Tnc* pro-fibrotic genes was significantly reduced as compared to vehicle treated animals (Fig. 6b). However, different from in vitro data, we did not observe a significant increase in cardiac *Vegfa* expression upon GSK-343 treatment (Supplementary Table 4), despite its bivalent gene status. This suggests that recruited myeloid cells might not be the principal source of *Vegfa* during the early steps of cardiac repair post-MI. In agreement with absence of major angiogenic or lymphangiogenic effects, both cardiac vessel density and lymphatic densities at 8 days post-MI were not significantly altered in GSK-343-treated groups as compared to vehicle controls (Supplementary Table 3 and Fig. S15).

By accelerating the resolution of the inflammatory phase, GSK-343 treatment may influence cardiac scar formation strength or kinetics. To investigate this, we evaluated cardiac function and morphology post-MI. Although infarct sizes were similar between the two groups at 3 days post-MI, we noticed a stabilization of infarct scar size, with less expansion by 8 days post-MI in GSK-343 as compared to vehicle-treated groups (Fig. 6d, e). In agreement with reduced ventricular wall stretching, due to less infarct expansion, GSK-343-treated mice displayed reduced cardiomyocyte hypertrophy at 8 days post-MI (Supplementary Table 3 and Fig. S15). Consistent with these findings we observed higher expression 8 days post-MI of both *Mb* and *Tnni3*, encoding respectively myoglobin and troponin I cardiomyocyte-specific genes in GSK-343-treated mice, suggesting limitation of cardiomyocyte cell death or dysfunction in the border zone (Fig. 6b). This is further confirmed by our observation of significantly reduced expression of *Nppb*, a well-known biomarker of deleterious cardiac remodeling and heart failure at 8 days post-MI. We conclude that Ezh2 inhibition with GSK-343 prevents infarct scar expansion through expedited inflammatory phase resolution, leading to increased scar maturation and potentially reduced cardiomyocyte remodeling or dysfunction. Indeed, we observed partial prevention of cardiac dysfunction 8 days post-MI, with reduced left ventricular (LV) dilation of both diastolic (Fig. 6f) and systolic (Fig. 6g) diameters, leading to

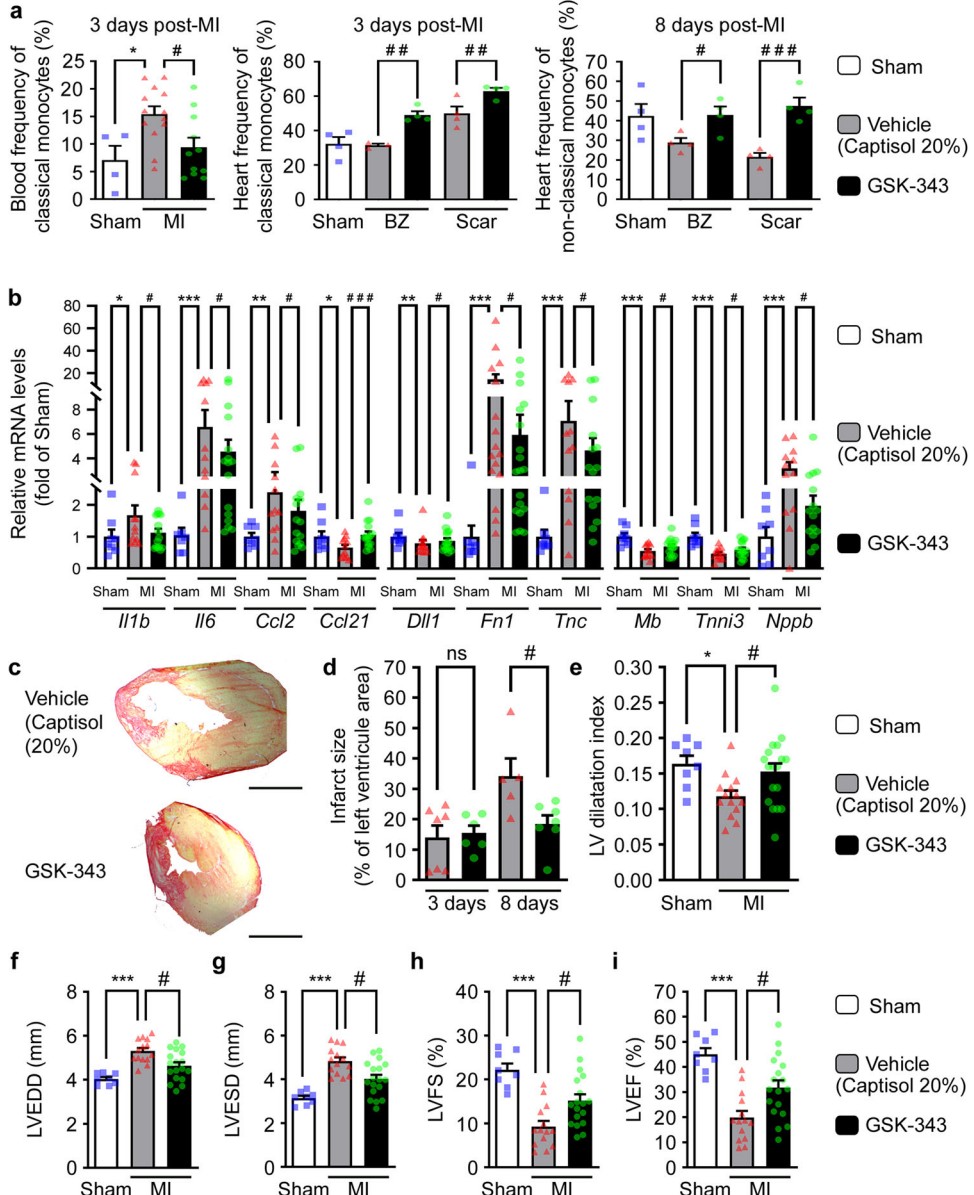

**Fig. 6 | Pharmacological Ezh2 inhibition with GSK-343 accelerates cardiac inflammatory resolution to prevent infarct expansion and subsequent cardiac remodeling and dysfunction post-MI.** Circulating (3 days post-MI) and cardiac (3 days and 8 days post-MI respectively for center and right panels) classical (Cd11b$^{hi}$Ly6c$^{hi}$) and non-classical (Cd11b$^{hi}$Ly6c$^{lo}$) monocyte population frequencies were evaluated by flow cytometry. Samples were collected from sham ($n = 4$) and MI mice treated daily either with vehicle (captisol 20%, $n = 13$ blood samples, $n = 4$ cardiac samples) or GSK-343 either from the infarcted scar area or from the healthy border zone (BZ). Data are expressed as mean frequency ± SEM of live CD45$^{+}$/Cd19$^{neg}$ cells. **b** Cardiac transcript levels of indicated genes measured by RT-qPCR from sham ($n = 8$), or vehicle ($n = 16$) or GSK-343-treated ($n = 18$) mice at 8 days post-MI. Values expressed as mean percentages of sham ± SEM with *B2m* serving as internal control. **c** Representative pictures of infarct scar and M-mode

echocardiography in sham, vehicle and GSK-343 treated mice at 8 days post-MI. Scale bar represents 1 mm. Infarct size assessed in vehicle ($n = 7$ and $n = 5$ respectively at 3 and 8 days post-MI) or GSK-343 treated ($n = 6$ and $n = 7$ respectively at 3 and 8 days post-MI) MI-mice (**d**). Data represent mean percentage of left ventricle area ± SEM. Quantitative analysis of LV dilatation index (**e**) and echo parameters for wall thickness: LVEDD (**f**), LVESD (**g**), LVFS (**h**) and LVEF (**i**) in sham ($n = 8$), vehicle ($n = 14$) and GSK-343 ($n = 18$) treated mice at 8 days post-MI. Data are presented as means ± SEM. For all panels (**a**–**i**), asterisk (*) and hashtag (#) symbols indicates statistically significant difference compared to sham and vehicle condition respectively after Kruskal–Wallis test. The *p* values are depicted as follows: *** and ### $p < 0.001$, ** and ## $p < 0.01$, * and # $p < 0.05$; ns: non-significant. Source data are provided as a Source Data file.

improved LV fractional shortening (Fig. 6h) and ejection fraction (Fig. 6i) after GSK-343 daily treatment. In contrast, none of these parameters were altered by GSK-343 at 3 days post-MI (Supplementary Table 5). We thus conclude that GSK-343 treatment accelerated inflammatory resolution, leading to improved infarct scar maturation, cardiomyocyte protection and subsequent reduction of infarct expansion, resulting in reduced cardiac remodeling and dysfunction post-MI.

## Discussion

We report, to the best of our knowledge, a new mechanism regulating monocyte and macrophage differentiation revealing cytoplasmic translocation of EZH2 as mediator of a cellular epigenetic switch facilitating M2-like immunomodulatory macrophage polarization through de-repressed expression of bivalent genes. This finding of cytoplasmic translocation of EZH2, while not reported previously in macrophages, has previously been described in T cells during actin

polymerization, where EZH2 was found to interact with Vav1 to regulate T-cell-receptor-mediated signaling[26]. Moreover, during megakaryopoiesis, Notch1 was described to induce EZH2 cytoplasmic translocation to mediate interactions with LIM domain kinase-1 (LIMK1), resulting in reduced Cofilin phosphorylation. This increases Cofilin activity and subsequently decreases filamentous actin content, preparing the cells for cell shape and size modifications required for megakaryocyte formation[27]. Finally, a recent report has indicated another cytoplasmic role for EZH2 linked to breast cancer metastasis. It was demonstrated that p38-induced phosphorylation of EZH2 at Tyr$_{367}$ promoted its cytoplasmic retention in breast cancer cells. This in turn enhanced EZH2 binding to Vinculin and other cytoskeletal regulators, leading to promotion of cell adhesion, migration, invasion and subsequent development of breast cancer metastasis[28]. Together with our findings, these data depict cytoplasmic roles of EZH2, reminiscent of our transcriptomic findings of improved homing and chemotaxis/ migration of M2 macrophages and GSK-343 treated monocytes. Interestingly, besides LIMK1 and Vav1, additional cellular partners of EZH2 have been identified that may participate to EZH2 cytoplasmic translocation. Among these putative partners, long noncoding RNA (lnc RNA) are interesting candidates. Indeed, EZH2 RNA Immune-Precipitation-sequencing (RIP-seq) experiments used to identify tissue-specific lnc RNA partners of EZH2 confirmed interaction with both Cardiac Hypertrophy Associated Epigenetics Regulator (CHAER)[29,30] and HOx Transcript Antisense RNA (HOTAIR) lnc RNA in both heart and blood[29]. Whereas Chaer was found to directly interact with PRC2, leading to inhibition of Ezh2-mediated repression of hypertrophic genes in the heart[30], Hotair interacted with PRC2 and enhanced EZH2 activity, leading to increased myofibroblast differentiation by promoting collagen and α-SMA expression. It remains to be determined whether these partners could be involved in regulation of Ezh2 cytoplasmic translocation during M2 immunomodulatory macrophage differentiation/polarization.

Although the data available on the role of EZH2 in the innate immune response remains scarce, a recent report examined the effects of Ezh2 deletion in macrophages and microglia. The authors reported that Ezh2 is required to repress Socs3 expression, necessary to promote pro-inflammatory gene expression in response to Toll-Like Receptor (TLR)-mediated signaling during macrophage and microglia activation in autoimmune inflammation[20]. This observation, in agreement with our data, suggests that EZH2 is a critical epigenetic check point regulator preventing immunomodulatory macrophage differentiation/polarization and contributing to poor resolution of cardiac inflammation post-MI. In this study, we show that pharmacological inhibition of EZH2 with GSK-343 promotes the capacity of human monocytes to accelerate cardiac inflammatory resolution through de-repression of bivalent genes. Further, we propose that this epigenetic control of bivalent gene expression is mediated by demethylation of the H3K27me3 repressive mark by epigenetic enzymes such as JMJD3 and UTX. Indeed, previous studies have shown that IL4 upregulates KDM6B (JMJD3) expression and activity in human monocytes. This in turn, decreases H3K27me3 levels at the IRF4 promoter, increases RNA polymerase II recruitment and promotes subsequent IRF4 mRNA expression. The implication of JMJD3 in bivalent genes re-expression was further confirmed by the inhibition of IRF4 expression after treatment with a pharmacological inhibitor of JMJD3, GSK-J4[12]. Moreover, recent publications from another group have revealed that the balance between the myeloid activity of Jmjd3 (Kdm6b)[31] and Ezh2[32] regulates atherosclerosis development. Even though the current literature seems to incriminate more particularly, JMJD3 (KDM6B), as the critical enzyme for M2 immunomodulatory macrophage phenotype differentiation; KDM6A and KDM6B expression data obtained in our laboratory from both murine and human differentiated M0, M1 and M2 macrophages in vitro, highlight the downregulation of KDM6A (UTX) in M1 macrophages compared to other macrophage subtypes

(Fig. S16). More studies, using for instance mouse model Kdm6a and/or Kdm6b specific deletion in myeloid cell lineage, are required to establish and discern the respective of JMJD3 and UTX in EZH2 inhibition-mediated bivalent genes upregulation. In our study, we observed a similar phenomenon (decrease of H3K27me3 level at IRF4 promoter enhancing its mRNA expression) after treatment of human monocytes with the EZH2 inhibitor GSK-343. This suggests that the balance between EZH2 and JMJD3 might be responsible for determining bivalent gene (IRF4, VEGFA and DLL1) promoter epigenetic mark H3K27me3 levels, which directly determines gene expression activity in monocytes during macrophage differentiation. Among these bivalent genes, DLL1 expression by endothelial cells has been shown to promote conversion of Ly6C$^{hi}$ classical monocytes into Ly6C$^{lo}$ alternative monocytes in vivo and in vitro[33]. We propose that DLL1 upregulation in monocytes could participate in cell-cell communication through NOTCH2 activation to stimulate inflammatory-to-immunoregulatory monocyte switch. We further postulate that EZH2 inhibition of immunomodulatory macrophage polarization is a global phenomenon, as this process occurred irrespective of M2 subtype and of patient categories. Our study is, however, limited by the lower number of patients included in our AMI cohort ($n = 7$) as compared to non-coronary patients. Moreover, we also provide, in vivo evidence of the protective effects induced by pharmacological EZH2 inhibition in a mouse MI model. Our data are consistent with a recent study highlighting increased cardiac EZH2 expression, as well as cardiac enrichment of known gene targets of EZH2, such as KLF15, in ischemic cardiomyopathy patients. Consequently, EZH2 has been proposed as a likely transcriptional regulator of cardiac gene expression in cardiovascular diseases[34] and more particularly ischemic cardiomyopathy[34,35]. In our mouse MI model, we observed accelerated inflammatory resolution upon EZH2 inhibition, in agreement with our in vitro data on altered gene expression and polarization of monocytes. Of note, we observed decreased cardiac expression upon GSK-343 treatment of both Il1b and Il6 pro-inflammatory cytokines and of Ccl2, implicated in cardiac monocyte recruitment and early immune response, as well as restored expression of Ccl21, involved in late immune response and cardiac repair[36]. We speculate that restored Ccl21 expression may relate to protection of cardiac lymphatics, as they constitute the main cardiac source of Ccl21. Moreover, our RNA-Seq data in human monocytes revealed that the CCL21 receptor, CCR7, is the among significantly upregulated genes after GSK-343 treatment. This indirectly suggests that GSK-343 may have enhanced both cardiac and lymphatic homing of myeloid cells, via promotion of Ccr7/Ccl21 signaling, resulting in accelerated clearance of cellular debris and inflammation in the heart[37] contributing to prevention of infarct scar expansion. Further, GSK-343 treatment may also have directly altered fibrosis and scar maturation, as suggested by the reduction in cardiac Fn1 and TnC expression as observed in mice post-MI. In addition to modulating collagen production, in part by enhancing TGFβ signaling, TnC has also been shown to act as a trigger for monocyte/macrophage recruitment[38] leading to accelerated adverse ventricular remodeling post-MI[39] and heart failure development. These immune effects of TnC seem to involve the suppression of Irf4 expression, necessary for M2 macrophage polarization. Taken together, our findings suggest that GSK-343-induced Irf4 upregulation in monocytes together with cardiac TnC downregulation could potentiate immunomodulatory macrophage polarization to accelerate cardiac repair and limit inflammation. Even though our data support the idea that myeloid Ezh2 inhibition with GSK-343 improves cardiac function post-MI, our study used global rather than specific myeloid cell targeting. Therefore, we cannot exclude direct beneficial impact of pharmacological Ezh2 inhibition also on other cell types. including cardiomyocytes, which may contribute at least in part to the functional cardiac effects. To address this question, we performed permanent left coronary artery ligation in myeloid-specific Ezh2 knockout mice (LysM-Cre$^{+/-}$ Ezh2$^{fl/fl}$) and control

littermates. Unfortunately, we failed to obtain adequate data and sufficient numbers of animals surviving MI to test this hypothesis (Fig. S17). Hence, future studies using other mouse models of Ezh2 deletion in the myeloid cell lineage (e.g., inducible Ezh2 myeloid deletion: *Cx3cr1-CreERT2*[+/-] *Ezh2*[fl/fl]) are required to discriminate whether cardiac functional improvement only dependents on the epigenetic switch in myeloid cells or also involved other cell types.

In conclusion, our study brings, for the first time, evidence of the key role of EZH2 as an epigenetic check-point regulator that prevents M2 immunomodulatory macrophage polarization. Promisingly, pharmacological EZH2 inhibition releases the repression of specific bivalent genes, resulting in increased expression of genes favoring M2-like immunomodulatory macrophage and monocyte polarization. In the setting of MI, the enhanced cardiac recruitment and activity of non-classical monocytes induced by EZH2 inhibition resulted in accelerated inflammatory resolution and decreased infarct scar expansion, leading to a reduction of cardiac remodeling and dysfunction post-MI. In conclusion, our data suggest EZH2 as an attractive therapeutic target to reduce cardiac inflammation and limit heart failure development following MI.

## Methods

### Human study design and approvals

Between Nov 2018 and June 2020, the EPICAM prospective study enrolled 48 patients at Rouen University Hospital. The CPP Ile de France V has approved the study august 10th 2018 (RCB: 2018-A02108-47). According to French legislation, all patients read the information note detailing the study protocol. All patients orally consented to the collection of an additional volume of blood during their usual care as well as the processing of their personal data (such as age, sex and coronary diagnosis) prior to the blood sampling. Inclusion criteria were: (1) age greater than or equal to 18 years and (2) admission to hospital for a coronary angiography. The exclusion criteria were (1) Infectious diseases, (2) Current pregnancy or breastfeeding, (3) Obesity (BMI > 30 kg/m²), (4) Hematological pathologies, (5) Anemia and (6) Inflammatory and autoimmune pathologies.

During the coronary angiography procedure, peripheral blood was collected into 4 BD Vacutainer® EDTA K2 tubes (Becton Dickinson Cat#367862). In-hospital data were entered into a dedicated database.

### Animal study design and approvals

All animal experiments performed in this study were approved by the regional ethics review board in line with E.U and French legislation, referred as APAFIS #8157-2016121311094625-v5 Normandy, APAFIS #31897-2021111911125883 v4 for mouse model of MI in vivo and as DUO #8980 for genetically modified mouse breeding, B6.129P2-Lyz2[tm1(cre)Ifo]/J (The Jackson Laboratory Cat#004781) and B6;129S1-Ezh2[tm2Sho]/J (The Jackson Laboratory Cat#022616). Commercial C57BL/6JRj (Janvier Labs) or genetically modified *LysM-Cre*[+/-] *Ezh2*[fl/fl] and *LysM-Cre*[-/-] *Ezh2*[fl/fl] female mice, 20–22 g (12–18 weeks), were maintained at the EU0477 animal care facility in ventilated cages with softly lit, environmental enrichment, ad libitum access to normal chow diet (Altromin International Cat#1310) and water (automated watering system), subjected to a standard 12-hour light-dark cycle at a constant temperature of 22 °C and a relative humidity kept between 45 and 65%. A two-week acclimatization period was observed for mice provided by Janvier Laboratories. We only used female mice for MI studies as they display lower mortality than male in the MI model, which help us reduce the numbers of animals included in our studies. In addition, females express both alleles of the H3K27me3 demethylase *Kdm6a* gene encoding the Utx protein. Mouse peripheral blood samples were collected in BD Vacutainer® EDTA K2 tubes (Becton Dickinson Cat#367862) from C57Bl/

6 J female mice aorta. After peripheral blood sample collection, mice were euthanized by cervical dislocation for tissue collection.

### Ezh2 myeloid deletion mouse model

*Lyz2*-Cre[+/-] (LysM-Cre[+/-], B6.129P2-Lyz2[tm1(cre)Ifo]/J, The Jackson Laboratory Cat#004781) and *Ezh2*[fl/fl] (B6;129S1-Ezh2[tm2Sho]/J, The Jackson Laboratory Cat#022616) murine breeders were generously given by Pr. Menno P. J. de Winther and Annette E. Neele from the department of Medical Biochemistry, Amsterdam Cardiovascular Sciences, Amsterdam Institute for Infection and Immunity, Amsterdam UMC, University of Amsterdam, Amsterdam, Netherlands. Myeloid cell-specific Ezh2 exons 16 and 17 deficient mice (*LysM-Cre*[+/-] *Ezh2*[fl/fl]) were obtained from a breeding with *LysM-Cre*[+/-] *Ezh2*[fl/fl] and *LysM-Cre*[-/-] *Ezh2*[fl/fl]. The genotype of each mouse was verified using the following primers after mouse ear punch genomic DNA extraction. Insertion of LoxP sites in the *Ezh2* gene was assessed by *Ezh2*[fl/fl] PCR using the following primers: *Ezh2*[fl/fl] *int17* forward primer: 5′-CATGTGCAGCTTTCTGTTCA-3′ and *Ezh2*[fl/fl] *int17* reverse primer: 5′-CACAGCCTTTCTGCTCACTG-3′ (WT 203 bp and flox ~300 bp amplicon). Knock-in insertion of Cre sequence within the promoter of the murine *Lyz2* gene was observed with the *Lyz2(LysM)-Cre* PCR using the following primers: *Lyz2-WT* forward primer the 5′-CTTGGGCTGCCAGAATTTCTC-3′ and *Lyz2-WT* reverse primer: 5′-TTACAGTCGGCCAGGCTGAC-3′ (346 bp amplicon); *Lyz2-Cre* forward primer: 5′-CTTGGGCTGCCAGAATTTCTC-3′ (*Lyz2-WT* forward primer) and *Lyz2-Cre* reverse primer: 5′-CCCAGAAATGCCAG ATTACG-3′ (700 bp amplicon). Finally, myeloid cells-specific *Ezh2 exons 16 and 17* deletion (*Ezh2LysM del ex16-17*) was visualized using the previously used flox primers sequence: *Ezh2*[fl/fl] int15 forward primer: 5′-CCCATGTTTAAGGGCATAGTG-3′ and *Ezh2*[fl/fl] *int17* reverse primer: 5′-CACAGCCTTTCTGCTCACTG-3′ (WT 2117 bp and *Ezh2del ex16-17* ~ 700 bp amplicon).

### Mouse model of Myocardial infarction

Left ventricular (LV) MI was induced in 20–22 g (12–18 weeks) C57BL/6JRj (Janvier Labs) or *LysM-Cre*[+/-] *Ezh2*[fl/fl] and *LysM-Cre*[-/-] *Ezh2*[fl/fl] female mice by permanent ligation of the left descending coronary artery as previously described[40]. GSK-343 (20 mg/kg, Interchim Cat# XLR94D) was dissolved in cyclodextrin-based Captisol® as a solubilization vehicle at 20% final concentration, as advised by the manufacturer to perform daily intraperitoneal injections starting from 6 h after surgery and daily until 3–7 days post-MI. Control mice received a vehicle solution (Captisol® 20%) injection of the same volume at the same time as treated mice.

### Cell isolation and culture

Monocytes were directly or indirectly (after red blood cell lysis (eBioscience Cat#00-4333-57) isolated by negative selection using EasySep™ monocyte isolation kits for human (StemCell Technologies Cat#19669) and mouse (StemCell Technologies Cat#19861) cells from fresh peripheral blood samples according to the manufacturer's instructions. TIB-204™ (WEHI-265.1) mouse monocyte cell line (ATCC Lot#4249478) was used in this study. All monocytes were cultured in Dulbecco's modified Eagle's medium (DMEM, Gibco, Cat# 41966-029) supplemented with 50 µM of 2-mercaptoethanol (Gibco Cat#21985-023), 10% Fetal Bovine Serum (FBS, Gibco Cat#10500-064) and Penicillin/Streptomycin (Sigma-Aldrich Cat#P4333) seeded at a cellular density >150 000 ¢/cm² for primary monocytes and >200,000 ¢/ml for TIB-204. M0 macrophages were differentiated from TIB-204 with 100 µM Phorbol 12-myristate 13-acetate (PMA, Sigma-Aldrich # P8139-1MG) and from selected primary monocytes using 50 ng/ml of either murine or human Macrophage-Colony Stimulating Factor (M-CSF) for 4 days. Further differentiation of M0 macrophages into polarized macrophages was obtained with 50 ng/ml LipoPolySaccharide (LPS, Sigma-Aldrich Cat#L6529) for M1 and with a combination of both IL4

and IL10 at a final concentration of 20 ng/ml respectively for M2 polarized macrophages for 2 days.

## Antibodies and reagents

For differentiation of primary monocytes into M0 macrophage in vitro, murine (Peprotech Cat#315-02) or human (StemCell Technologies #78057.1) M-CSF was used. The subsequent M0 into M2 macrophage polarization was obtained using a cocktail of murine Il4 (Peprotech Cat#214-14) and Il10 (Peprotech Cat#210-10) or human IL4 (StemCell Technologies Cat#78045.1) and IL10 (StemCell Technologies Cat#78024.1). M2 macrophage subpopulation phenotypes were obtained after incubation with 20 ng/ml of murine Il4 (Peprotech Cat#214-14), Il10 (Peprotech Cat#210-10), and Tgfb1 (R&D Sytems Cat#7666-MB-005) alone or in combination depending on the M2 macrophage subtype desired. The Ezh2 inhibitor GSK-343 (Sigma-Aldrich Cat#SML0766) was dissolved in dimethyl sulfoxide (DMSO, Sigma-Aldrich Cat#D1435) and used at a final concentration of 5 µM for in vitro experiments.

The following antibodies were used at the indicated dilutions:

Immunohistochemistry: rat anti-CD11b (BD Pharmingen Cat#557395, 1:1500), rat anti-CD68 (eBiosience Cat#14-0681, 1:800), biotinylated rat anti-CD68 (FA_11, Miltenyi Biotec Cat#130-102-025, 1:200), biotinylated rat anti-F4/80 (BM9 Abcam Cat#15694, 1:75), goat anti-CD206 (Thermo Fisher Cat#PA5-46994, 1:1500), rabbit anti-EZH2 (D2C9, Cell Signaling Cat#5246, 1:1000), rat anti-iNOS (W16030C, Biolegend Cat#696802, 1:500), WGA-FITC (Interchim#FP-CE8070, 1:100), biotinylated rabbit anti-Lyve1 (eBioscience#13-0443, 1:400), biotinylated rat anti-CD31 (BD pharmingen#553371, 1:50); biotinylated rabbit anti-Cx3cr1 (BIOSS Cat#bs-1728R-Biotin, 1:100), rat anti-Ly6c (W16030C, Abcam Cat#ab15627, 1:400), rat anti-Trem2 (RM0139-5J46, Abcam Cat#ab86491, 1:100), rat anti-Cd86 (B87-2, eBiosience Cat#14-0862-82, 1:100), goat anti-IL-1 beta /IL-1F2 (R&D Systems Cat#AF401-NA, 1:100) and rat anti-MHC Class II (I-A_I-E) (M5/114.15.2, eBiosience Cat#14-5321-82, 1:200). Donkey anti-species fluorescent secondary antibodies and probes were used at indicated dilution: donkey anti-rat-AF488 (Jackson ImmunoResearch Cat#712-545-153, 1:400), donkey anti-rat-Cy3 (Jackson ImmunoResearch Cat#712-166-153, 1:400), donkey anti-rabbit-Cy3 (Jackson ImmunoResearch Cat#711-165-152, 1:400), donkey anti-rabbit-AF647 (Jackson ImmunoResearch Cat#711-605-152, 1:400), donkey anti-rat-AF647 (Jackson ImmunoResearch Cat#712-605-153, 1:400), donkey anti-goat-biotin (Jackson ImmunoResearch Cat#705-065-147, 1:1000), streptavidin-Cy5 (GE Healthcare Amersham Cat#PA 45001, 1:1500). Nuclei were stained with DAPI (Vectashield Cat#H1200).

Chromatin Immunoprecipitation: 2 µg of the following antibodies were used: rabbit anti-H3K4me3 (Millipore Cat#07-473), rabbit anti-H3K27me3 (Millipore Cat#07-449), or normal rabbit IgG (Millipore Cat#12-370).

Flow cytometry: anti-mouse CD45 (30F11)-BV711 (Biolegend Cat#103147, 1:200); anti-mouse CD11b (M1/70)-FITC (BD Pharmingen Cat#553310, 1:200); anti-mouse Ly6C (HK1-4)-APC/Cy7 (Sony Cat#1240130, 1:200); anti-mouse Cx3cr1 (SA011F11)-BV421 (Biolegend Cat# 149023, 1:200); anti-mouse CD86 (GL-1)-BV650 (Sony Cat#1125175, 1:200); anti-mouse CD3ε (17A2)-BV785 (Biolegend Cat#100355, 1:200); anti-mouse CD115 (AFS798)-PE (Sony Cat#1277530, 1:200); anti-mouse CD11c (N418)-PE/Texas Red (Sony Cat#1186740, 1:200); anti-mouse CD19 (6D5)-PE/Cy5 (Biolegend Cat#115509, 1:200); anti-mouse CD206 (C068C2)-APC (Sony Cat#1308540, 1:200); anti-mouse IA-IEk (MHCII) (M5/114.15.2)-Alexa Fluor 700 (Sony Cat#1138110, 1:200); anti-mouse F4-80(BM8)-BV605 (Sony Cat#1215665, 1:200), LIVE/DEAD Viability/Cytotoxicity-UV (Invitrogen Cat#L23105, 1/1000).

Western blotting: rabbit anti-pan histone H3 (Millipore Cat#07-690, 1:100,000), rabbit anti-H3K27me3 (Millipore Cat#07-449, 1:2000), polyclonal goat anti-rabbit-HRP (Dako Cat#P0448, 1:10,000).

## Immunohistochemistry analysis

Frozen OCT-embedded mouse hearts were sectioned on a cryostat with a section thickness of 8 µm. Air dried sections were fixed in ice cold acetone (−20 °C) for 10 min before a 1 h permeabilization step in PBS containing 0.1% saponin; 1% BSA; 4% donkey serum. Incubation with desired primary antibody was performed 1 h to overnight. Primary antibody signal was revealed using donkey anti-species fluorescent secondary antibody. Pictures were taken on Zeiss microscope imager Z.1 equipped with an apotome.

## Immunocytochemistry analysis

TIB-204 and primary mouse monocytes were stained in suspension while adherent macrophages were directly stained on 14 mm ø coverslips. Briefly, cells were fixed for 10 min in 1 or 4% paraformaldehyde respectively for suspension or adherent cells. Permeabilization was then performed using PBS containing 0.25% Triton X-100 or 0.1% saponin respectively for suspension or adherent cells. Cells were incubated overnight with desired primary antibodies, as described above, at indicated concentration after 30 min incubation in PBS containing 0.1% saponin; 1% BSA; 4% donkey serum.

## RNA extraction

Total RNA was extracted from at least $3 \times 10^6$ human and $5 \times 10^5$ murine monocytes treated or not with 5 µM GSK-343 for 72 h using NucleoSpin RNA XS kit (Macherley-Nagel Cat# 740902) for in vitro studies or from 5 mg of cardiac tissue using RNeasy Mini Kit (Qiagen Cat#74104) including a genomic DNA digestion step with DNAse I (Qiagen Cat#79254).

## Gene expression profiling by RNA-Sequencing

For RNAseq, we used total RNA extracted from three independent non-coronary patients treated with 5 µM GSK-343 or vehicle for 72 h therefore representing three biological replicates. Total RNA was sent for library preparation, mRNA sequencing and bio-informatics analysis to Novogene Company (Novogene company Ltd, Cambridge, UK). 400 ng of total RNA with a concentration >20 ng/ml and a RIN ratio >8 were used for Poly-A mRNA-seq, sequencing length of 150 nt paired-end (PE150) with the Illumina NovaSeq 6000. Reads were aligned to the human genome build hg38 and the transcript assembly GRCh38 from Ensmbl using Hisat2 version 2.0.5[41]. Transcript quantification was performed with featureCounts from the subread package[42]. Differential analysis, including normalization, was performed with DESeq2 bioconductor package version 1.22.1[43] using R version 3.5.1. Genes significantly up- or downregulated upon epigenetic combined drug treatment (adjusted $p$-value < 0.05) were submitted to DAVID Gene Ontology Analysis version 6.7[44,45]. To generate heat maps, expression values in FPKM for genes within the selected GO Categories were retrieved. Heat maps were generated with Cluster version 3.0[46] and JavaTree View version 1.1.6r4[47]. The RNA-Seq data generated in this study have been deposited in the public Gene Expression Omnibus (GEO) database under accession code GSE165543.

## Reverse transcription-quantitative polymerase chain reaction (RT-qPCR)

RT-qPCR was performed with 0.4–1.5 µg total RNA isolated as previously described above. Extracted RNA was annealed with 1875 ng random primers (Invitrogen, Cat#48190-011) for 5 min at 65 °C followed by holding step at 4 °C. Reverse transcription step was performed using 200 U M-MLV reverse transcriptase (Invitrogen, Cat#28025-013) mixed with 30 U RNAse OUT (Invitrogen, Cat#10777-019), M-MLV reverse transcriptase reaction buffer (Invitrogen, Cat#18057-018) and 1 mM final dNTP mix (Invitrogen, Cat#10297-018). Complementary DNA (cDNA) synthesis was carried out at 37 °C for 1 h followed by 95 °C for 5 min cycle and final hold at 4 °C. cDNA was diluted 1:2 or 1:4 in nucleases free water before amplification with

LightCycler®480 SYBR Green I Master (Roche, Cat#04-707-516-001) using Light Cycler 480 real-time PCR system (Roche). Absolute abundance of genes relative to *B2M* housekeeping gene expression was calculated based on a cDNA standard curve using Light Cycler 480 Software 1.7 (Roche). A complete list of primer sequences is provided as supplementary data 4 and 5.

## ChIP-seq data analysis

We downloaded data sets from the ENCODE portal[48] (https://www.encodeproject.org/) with the following identifiers: ENCSR267NWZ, and ENCSR000ASK respectively for human CD14[+] monocytes H3K4me3 (GSM1003536) and H3K27me3 (GSM1003564) ChIP-seq data sets[23,24].

Reads were aligned to the hg19 genome build using Bowtie2 version 2.3.4.1[49,50]. Unmapped and duplicate reads were removed with Samtools version 1.5[51]. Peak calling was performed with MACS2[52,53] using default parameters for H3K4me3 samples and with the addition −borad option with H3K27me3 samples. Overlapping peaks were identified with the intersect Bed tool from the bedtools package version 2.26[54]. Annotation of common H3K4me3 and H3K27me3 peaks was performed with PAVIS[55]. The H3K4me3 and H3K27me3 ChIP-Seq peak calling data used in this study are available in the public GEO database under accession code GSE226811.

## Native chromatin immunoprecipitation (NChIP)

H3K4me3 and H3K27me3 histone modification levels were measured by native ChIP (NChIP) as described[56]. ChIPed DNA was purified by phenol-chloroform extraction, precipitated with ethanol and specific DNA sequences were amplified with LightCycler®480 SYBR Green I Master (Roche, Cat#04-707-516-001) using Light Cycler 480 real-time PCR system (Roche). ChIPed DNA quantity was calculated compared to a genomic DNA standard curve using Light Cycler 480 Software 1.7 (Roche). A complete list of primer sequences is provided as Supplementary Data 6.

## Heart dissociation

Cardiac samples were collected at 3 or 8 days post-MI. Left ventricles (LV) were harvested after perfusion with physiologic serum at 37 °C. Infarcted Scar and Border Zones (BZ) were carefully separated and collected in 50 ml-falcon tube containing 5 ml of RPMI 1640 medium. Three LVs from same group were collected per tube. Digestion enzymes cocktail (5 mg of collagenase II (Gibco Cat#17101015), 6 mg of Dispase (Sigma-Aldrich Cat#D4693), 300 µg of DNAse I (Sigma-Aldrich Cat#DN25) per 5 ml) was added and sample were dissociated through GentleMACS™ (Miltenyi biotec) for 15 min. After dissociation, samples were filtered through 70 µm and then 40 µm cell strainer and prepared for FACS staining in PBS-2% FBS buffer.

## Western blotting analysis of histone post-translational modifications

TIB-204 nuclei were lysed using Triton Extraction Buffer (TEB: PBS containing 0.5% Triton X 100 (v/v); 2 mM phenylmethylsulfonyl fluoride (PMSF), 0.02% (w/v) NaN3) supplemented with protease inhibitor cocktail before overnight histones acid extraction in 0.2 N HCl at 4 °C. Total amount of histones was measured using BioRad protein Assay (Biorad, Cat#500-0006). 2 µg of total histones were resolved by SDS-PAGE electrophoresis followed by transfer on nitrocellulose membrane before hybridation with desired primary antibody previously described overnight at 4 °C. Primary antibody signal was amplified using horseradish peroxidase (HRP) coupled specie corresponding secondary antibody.

## Cardiac inflammation analysis by flow cytometry analysis

Cardiac immune cell infiltration was assessed by flow cytometry using LSRFortessa Cell Analyzer (BD Bioscience) and FlowJo v10.8.1 for post-acquisition analysis. CD45[+] leucocytes gating strategy was used to exclude endothelial cells and other cardiac cell types (Fig. S14). On live cells, lymphocytes were gated out by CD3 or CD19 staining. Then, dendritic cells were excluded from the analysis by gating out CD11c[+] cells. In the CD11b[+]CD11c[−] pool, monocyte/macrophage cell populations were analyzed based on their expression of Ly6C versus Cx3cr1. Granulocytes were excluded by their lower expression of Cx3cr1 and higher Side Scatter (SSC) profile compared to monocytes/macrophages subset. Classical/inflammatory and non-classical monocytes/macrophages from peripheral blood (Fig. S14a) and heart tissue (Fig. S14b) samples were defined as Ly6C[hi]/Cx3cr1[hi] and Ly6C[lo]/Cx3cr1[hi], respectively.

## Cardiac functional analysis

Non-invasive echocardiography was performed as described[57] 3- and 7-days post-MI and GSK-343 treatment. Briefly, mice were anesthetized using Isovet - 2% (Osalia, Cat#3248850) and echography was performed with VEVO 3100 ultrasound echograph equipped with a MX550D probe (Fujifilm VisualSonics Cat#51073-45). Short axis view of the left ventricular was performed at the level of papillary muscle and M-mode tracing was recorded. Doppler was used to calculate Velocity Time Integral (VTI) in pulmonary artery. Left ventricular diameter was measured by image analysis using software VEVOLAB V3.1.1.

## Statistical analyses

Data are obtained from at least three independent experiments each in duplicate or triplicate for in vitro studies. Number of individual animals in each group is indicated for in vivo data. Data are expressed as mean values or percentages of control values ± SEM. When indicated, statistical significance was determined by non-parametric Kruskal−Wallis test followed by Dunns post-hoc or two-way ANOVA with Sidak's multiple comparisons test using Graphpad Prism 8 software. The use of non-parametric Kruskal−Wallis test was determined depending Shapiro-Wilk normality test data distribution. Data are considered to be significantly different at values $p < 0.05$. The $p$ values are depicted as asterisks (*) or hashtag (#) symbols in the figures as follows: *** or ### $p < 0.001$; ** or ## $p < 0.01$; * or # $p < 0.05$; ns: non-significant.

## Reporting summary

Further information on research design is available in the Nature Portfolio Reporting Summary linked to this article.

## Data availability

Source data are provided with this paper. The RNA-Seq and ChIP-seq data generated in this study have been deposited in the public GEO database under accession code GSE165543 and GSE226811 respectively: GSE165543 and GSE226811. Source data are provided with this paper.

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

## Acknowledgements

We thank all physicians from the department of cardiology (Rouen University Hospital) for their help in the collection of peripheral blood samples from patients; Nicolas Perzo, Dr. Thomas Duflot, and Pr. Jérémy Bellien for critical commenting on experiments and Gaëtan Riou from CyFlow flow cytometry and cell analysis facility platform (Institute for Research and Innovation in Biomedicine, IRIB) for his help in performing flow cytometry experiments and analysis. We thank the ENCODE Consortium and the Bradley Bernstein, Broad Institute of MIT and Harvard ENCODE production laboratory for generating the GSE29611 ChIP-seq data sets, and more particularly for generating respectively GSM1003536 and GSM1003564 human CD14+monocytes H3K4me3 and H3K27me3 ChIP-seq data sets. Professors E.D. (MD-PhD), V.R. (MD-PhD), D.G. (MD-PhD); Doctors J.R. (PhD), D.G. (PhD), V.T. (PhD), D.B.G. (PhD), E.B. (PhD), S.F. (PhD); Mrs S.R. (BsC), A.D. (BsC), L.D.M. (MsC), M.V. (BsC), Z.B. (BsC), C.V. (MsC) and Mr T.L. (MsC), J.P.H. (BsC) are all part of the University Hospital Federations CArdiac Research Network on Aortic VAlve and heart faiLure (FHU CARNAVAL) consortium for which a GCS G4 grant is granted. This study was supported with grants from the University of Rouen Normandy, the GCS G4 FHU Early Markers of Cardiovascular Remodeling in Valvulopathy and Heart Failure (FHU REMOD-VHF) and FHU CARNAVAL as well as generalized institutional funds (INSERM U1096 EnVi laboratory) from French National Institute of Health and Medical Research (INSERM) and the Normandy Region together with the European Union. Julie Rondeaux is co-supported by a fellowship from European Union and Région Normandie. Europe gets involved in Normandie with European Regional Development Fund (ERDF): CPER/FEDER 2015 (DO-IT) and CPER/FEDER 2016 (PACT-CBS). This project required the use of equipment acquired by the Hospital-University Research in Health project Search Treatment and improve Outcome for Patients with Aortic Stenosis (RHU STOP-AS) supported by the French Government and managed by the National Research Agency (ANR) under the program "Investissements d'avenir" with the reference ANR-16-RHUS-0003. Figure S17 was partly generated using Servier Medical Art, provided by Servier, licensed under a Creative Commons Attribution 3.0 unported license.

## Author contributions

J.R. and S.F. designed, performed, and analyzed all in vivo and in vitro experiments. D.G. performed and analyzed echocardiography in mice. S.R. designed and carried out gene expression experiments. V.T. and T.L. performed and analyzed flow cytometry. A.D. performed and analyzed immunohistochemistry and histology. A.C. and M.B. performed ChIP-seq and RNA-seq analysis. L.D.M. produced and genotyped LysM-Cre−/− Ezh2fl/fl and LysM-Cre+/− Ezh2fl/fl mice and participated to the preparation of flow cytometry samples. J.P.H. and M.V. carried-out mouse MI model. A.N. and MdW gave LysM-Cre−/− Ezh2fl/fl and LysM-Cre+/− Ezh2fl/fl Ezh2 fl/fl murine breeders and participated to the reviewing process. Z.B. and C.V. coordinated collection of peripheral blood samples from cardiology patients. D.B.G. and E.D. designed and coordinated collection of peripheral blood samples from cardiology patients (project EPICAM n°18.07.19.36852 CAT 3, n°-RCB: 2018-A02108-47). D.G. and V.R. supervised experiments and provided experimental advice. E.B. participated in the design of in vivo studies. S.F. designed, supervised, and participated in all in vivo and in vitro experiments. The article draft was prepared by J.R., S.F., and E.B. All of the data collected for the review process were analyzed by S.F. S.F. and E.B. wrote the revised version of the manuscript. All authors approved the final revised version of the article.

## Competing interests

The authors declare no competing interests.
