## [Peer Review File · Nature Communications]

Ezh2 emerges as an epigenetic checkpoint regulator during monocyte differentiation limiting cardiac dysfunction post-MIREVIEWER COMMENTS

Reviewer #1 (Remarks to the Author):

This is a well written paper that details the role of histone H3K27 methylation in M2-like macrophage polarization after Myocardial Infarction (MI). They found that EZH2, responsible for H3K27 methylation, plays a role during monocyte differentiation in vitro as well as in M2 macrophages in vivo during post-MI cardiac inflammation. The study is important and provides novel insight into the role of EZH2, however, EZH2 often works in concert with JMJD3 and UTX and the effects of these demethylases is not examined. I have the following comments:

1. The definitions of M1/M2 are antiquated. It is widely considered that there is a spectrum of phenotypes that exist in tissues and thus, M1/M2 are used for in vitro assessment only. Additionally, it would be useful in Figure 1 to examine multiple time points post ligation.
2. In Figure 2 it is unclear why F4/80 - a resident tissue marker – was used. Further, Ly6C should be subdivided into low vs high as these represent different populations in the monocytes that differentiate into either inflammatory or alternative macrophages.
3. Information on the donors used in Figure 4 is necessary.
4. EZH2 inhibition in the in vivo should be done and examination of macrophages (similar to figure 1) would give additional information on how this influences in vivo phenotypes.
5. The role of JMJD3 and UTX inhibition on H3K27 should be examined in this model. The data presented that JMJD3 promotes an M2 phenotype is dated as there is overwhelming literature that JMJD3 can promote a macrophage inflammatory phenotype via promoting NFkB transcription – hence, the concept that it skews to M2 is not consistent with current literature – the role of these enzymes in this complex are much more complex and regulate both M1 and M2 genes – the timing in the in vivo is relevant, hence seeing more time points, during the initial injury and during the repair phases is important.

Reviewer #2 (Remarks to the Author):

This interesting manuscript examines epigenetic regulation of myeloid cells in the setting of infarct healing, specifically EZH2, responsible for H3K27 methylation. After MI and in vitro, EZH2 localizes in the cytoplasm of infarct macrophages, rendering it inactive.

Pharmacological EZH2 inhibition reduced H3K27 methylation, affected macrophages gene expression and phenotypes, and improved post MI recovery. A number of in vitro studies were done with human monocytes, including from infarct patients, that were differentiated in vitro.

While interesting and potentially important, the study has some shortcomings. I can see 2 major limitations:

The data do not mechanistically show that EZH2 manipulation acts via macrophage phenotypes. While this is not unlikely, it isn't proven. This would be best accomplished with in vivo studies that delete EZH2 specifically in macrophages.

The second major limitation is the overuse of the M1/M2 nomenclature and reliance on in vitro data. Admittedly, the M1/M2 nomenclature is valid for in vitro experiments, but does not really reproduce in the infarct. Studies have shown that certain M1/M2 genes are regulated in parallel, and that the macrophage subsets or phenotypes are more complex than what is seen on plastic, which lacks the typical microenvironment.

Minor comments:

Fig. 1: macrophages are often autofluorescent. Please provide controls to exclude this is the case.

Fig. 3b and 4e: FDR values would be more appropriate than p values

Fig. 6: please provide FACS plots to illustrate gating

Fig. S4: differences may become significant in larger cohorts (only 7 infarct patients). However

this is a minor point since these data are not novel.

Supplemental Table 5: Please provide example histology images for the respective stains and groups

We sincerely thank the two reviewers, for providing most helpful and insightful comments on our manuscript and for giving us the opportunity to revise our study to address the reviewers questions and suggestions. During this revision, we have carried out additional experiments to generate new data, which has been included in our new manuscript (3 new panels in Figure 1, modified panel in Figures 2; 3, 4, 5 and 6, 9 new supplemental figures and 1 supplemental table).

We have refined our previous data by :

- Better characterizing of macrophage subtypes and quantifying Ezh2 cellular localization in all of these macrophage subsets at several time points post-MI (sham, 24h, 72h and 7 days) *in vivo* (reviewer #1 question 1 and #2 major comment 2, new Figure 1 panels c, d and g as well as new supplementary Figures S1 to S6)
- Providing additional support for the notion that Ezh2 inhibition in macrophages is responsible of the observed benefit on cardiac function after MI (in a mouse model of Ezh2-specific deletion in macrophages: LysM-Cre^{+/-} Ezh2 fl/fl) in this response to the reviewers' comments (Reviewer #2 major comment 1)
- Providing i) a new non adherent and adherent monocyte consistent staining (reviewer #1 question 2, new Figure 2 panels a and b and new supplementary Figure S7); ii) more information about the patient characteristics (reviewer #1 question 3, new supplementary table 2); iii) Flow cytometry gating strategy (reviewer #2 minor point 3, new supplementary Figure S14); iv) examples of Immunohistochemistry stains and auto-fluorescence negative controls (reviewer #2 minor points 1 and 5, new supplementary Figures S1 to S6 and S15); Benjamini p-values for GO Biological processes (reviewer #2 minor point 2, new Figure 4 panel e)

In all the figures, we have replaced our bar graphs with plots that feature information about the distribution of the underlying data. Finally, we have carefully rearranged both the manuscript and figures, according to the reviewers' remarks and suggestions, in order to improve the fluidity of our data including the modification of the M1/M2 nomenclature for macrophage subpopulations *in vivo* and *ex vivo* (reviewer #2 question b).

We provide beneath our point-by-point response to the reviewers' comments (high-lighted in blue).

Reviewer #1 (Remarks to the Author):

This is a well written paper that details the role of histone H3K27 methylation in M2-like macrophage polarization after Myocardial Infarction (MI). They found that EZH2, responsible for H3K27 methylation, plays a role during monocyte differentiation *in vitro* as well as in M2 macrophages *in vivo* during post-MI cardiac inflammation. The study is important and provides novel insight into the role of EZH2, however, EZH2 often works in concert with JMJD3 and UTX and the effects of these demethylases is not examined.

We deeply thank the reviewer for laudatory comments regarding our study. We agree that EZH2 often works in concert with JMJD3 and UTX and provide an answer to this comment below (see comment 5).

I have the following comments:

1. The definitions of M1/M2 are antiquated. It is widely considered that there is a spectrum of phenotypes that exist in tissues and thus, M1/M2 are used for *in vitro* assessment only. Additionally, it would be useful in Figure 1 to examine multiple time points post ligation.

We thank the reviewer for this very relevant comment regarding M1/M2 denomination for *in vivo* macrophage subsets and agree with the reviewer's concern. We therefore modified the denomination of *in vivo* macrophage phenotypes to reflect the expression of their pro-inflammatory or immunomodulatory status regarding the combination of characteristic markers (*i.e.* Cd68⁺/iNos⁺, Cd68⁺/Cd86⁺, Cd68⁺/MHCII⁺, Cd68⁺/II1b⁺ for pro-inflammatory macrophages and Cd68⁺/Cd206⁺, Cd68⁺/Trem2⁺, Cd68⁺/Lyve1⁺ for immunomodulatory macrophages).

To answer the comment of the reviewer, we have performed new immunohistochemistry stainings to better characterize *in vivo* macrophage phenotypes (*i.e.* Cd68⁺/Cd86⁺, Cd68⁺/MHCII⁺, Cd68⁺/II1b⁺ for pro-inflammatory macrophages and Cd68⁺/Trem2⁺, Cd68⁺/Lyve1⁺ for immunomodulatory macrophages) as depicted in new Figure 1 panels c, d and g. Moreover, we have quantified the correlation between Ezh2 subcellular localization and either the pro-inflammatory or the immunomodulatory phenotype of each macrophage subset considered (*i.e.* Cd68⁺/Cd86⁺ in supplementary Figure S1, Cd68⁺/MHCII⁺ in supplementary Figure S2, Cd68⁺/II1b⁺ in supplementary Figure S3, for pro-inflammatory macrophages and in Cd68⁺/Cd206⁺ supplementary Figure S4, Cd68⁺/Trem2⁺ in supplementary Figure S5 and Cd68⁺/Lyve1⁺ in supplementary Figure S6 for immunomodulatory macrophages) in Sham mice (to characterize resident macrophage populations) and at 3 time points post-MI (one early time point: 24h, one intermediate time point: 72h and a late time point: 7 days) corresponding to critical time points previously described during post-MI inflammatory kinetics¹⁻³.

As expected from our previous studies, we observed that the mouse healthy heart is widely populated with cardiac resident macrophages (Cd68⁺/Cd86⁻ in supplementary Figure S1, Cd68⁺/MHCII⁻ in supplementary Figure S2, Cd68⁺/Il1b⁻ in supplementary Figure S3, Cd68⁺/Cd206⁺ supplementary Figure S4, Cd68⁺/Trem2⁺ in supplementary Figure S5 and Cd68⁺/Lyve1⁺ in supplementary Figure S6). Permanent coronary ligation induces recruitment of pro-inflammatory macrophages (Cd68⁺/Cd86⁺ in supplementary Figure S1, Cd68⁺/MHCII⁺ in supplementary Figure S2, Cd68⁺/Il1b⁺ in supplementary Figure S3, Cd68⁺/Cd206⁻ supplementary Figure S4 and Cd68⁺/Lyve1⁻ in supplementary Figure S6) at an early time point post-MI (24h). This pro-inflammatory subset is gradually replaced by the immunomodulatory macrophage subset (Cd68⁺/Cd86⁻ in supplementary Figure S1, Cd68⁺/MHCII⁻ in supplementary Figure S2, Cd68⁺/Il1b⁻ in supplementary Figure S3, Cd68⁺/Cd206⁺ supplementary Figure S4, Cd68⁺/Trem2⁺ in supplementary Figure S5 and Cd68⁺/Lyve1⁺ in supplementary Figure S6) from intermediate (72h) to late (7 days) time points after MI.

Interestingly, and in agreement with our previous observations, we confirmed that Ezh2 is predominantly located in the nucleus of pro-inflammatory monocytes (Figure 1a), and macrophages (Cd68⁺/iNos⁺ in Figure 1b, Cd68⁺/Cd86⁺ in Figure 1c and supplementary Figure S1, Cd68⁺/MHCII⁺ in Figure 1d and supplementary Figure S2, Cd68⁺/Il1b⁺ in supplementary Figure S3, Cd68⁺/Cd206⁻ Figure 1e and supplementary Figure S4 and Cd68⁺/Lyve1⁻ in supplementary Figure S6) at 24h post-MI. In contrast Ezh2 is located in the cytoplasm of immunomodulatory (both resident in sham mice and differentiated macrophages mostly from 72h to 7 days post-MI) macrophages (Cd68⁺/iNos⁻ in Figure 1b, Cd68⁺/Cd86⁻ in Figure 1c and supplementary Figure S1, Cd68⁺/MHCII⁻ in Figure 1d and supplementary Figure S2, Cd68⁺/Il1b⁻ in supplementary Figure S3, Cd68⁺/Cd206⁺ Figure 1e, f and supplementary Figure S4 and Cd68⁺/Lyve1⁺ in Figure 1g and supplementary Figure S6). These data strongly support our initial observations of an ectopic localization of Ezh2 in immunomodulatory macrophage phenotypes within the heart, independently of their resident or infiltration status.

2. In Figure 2 it is unclear why F4/80 - a resident tissue marker – was used. Further, Ly6C should be subdivided into low vs high as these represent different populations in the monocytes that differentiate into either inflammatory or alternative macrophages.

We agree with the reviewer's comment pointing out this inconsistency in the cellular markers used to characterize non-adherent (previous Figure 2a, new supplementary Figure S7a) and adherent (previous Figure 2b, new supplementary Figure S7b) mouse monocytes *in vitro*. In the revised manuscript, we have now stained non-adherent (new Figure 2a) and adherent (new Figure 2b) mouse monocytes with Cd68 to gain consistency in the choice of a characteristic macrophage marker and with Cd11b as well-known monocyte marker. We agree with the reviewer's comment about the subdivision into low and high Ly6c monocytes further differentiating into alternative versus inflammatory macrophages. Regarding to this comment, we (new supplementary Figure S7a) and

the manufacturer (StemCell Technologies in 2 posters titled “A simple new method for negative enrichment of mouse blood and bone marrow” and “A rapid new procedure for negative enrichment of monocytes from mouse blood and bone marrow”) confirm that the EasySep™ mouse monocyte isolation kit (StemCell Technologies Cat#19861) used to negatively isolate cells from fresh peripheral blood does not discriminate Ly6c^{hi} and Ly6c^{lo} monocytes. We believe that this Ly6c non-discriminating selection protocol is a strength for our study preventing bias through inadvertent enrichment of one or the other of these cell subpopulations. As depicted in supplementary S7a, we found that Ezh2 is located in the nucleus of both Ly6c^{hi} inflammatory and Ly6c^{lo} alternative monocytes.

3. Information on the donors used in Figure 4 is necessary.

We apologize for forgetting to provide information about the 3 donors used in our RNA-sequencing. This has been added in the revised manuscript in supplementary 2

4. EZH2 inhibition in the *in vivo* should be done and examination of macrophages (similar to figure 1) would give additional information on how this influences *in vivo* phenotypes.

We thank the reviewer for this comment. We have evaluated the phenotype of macrophages upon EZH2 inhibition 3 and 8 days after MI. However, we have not been able to find a significant difference regarding the macrophage phenotype by immunohistochemistry (new supplementary Figure S15 and supplementary table 6). Nevertheless, flow cytometry analysis performed on mice hearts after GSK-343 treatment 3 and 8 days post-MI (Figure 6a center and right panels) indicates that cardiac inflammatory kinetic is accelerated with EZH2 inhibition. Indeed, we observed improved recruitment of classical monocytes (pro-inflammatory macrophage precursors) 3 days post-MI (Figure 6a center panel) and of non-classical monocytes (immunomodulatory macrophage precursors) 8 days post-MI (Figure 6a right panel) upon EZH2 inhibition. Interestingly, we observed that EZH2 inhibition reduces the expression of *Ccl2* while normalizing the expression of *Ccl21* immunoregulatory cytokine (Figure 6b) arguing for an acceleration of the cardiac inflammatory kinetics⁴⁻⁶ and thus of the macrophage switch from pro-inflammatory toward immunomodulatory phenotypes.

5. The role of JMJD3 and UTX inhibition on H3K27 should be examined in this model. The data presented that JMJD3 promotes an M2 phenotype is dated as there is overwhelming literature that JMJD3 can promote a macrophage inflammatory phenotype via promoting NFkB transcription – hence, the concept that it skews to M2 is not consistent with current literature – the role of these enzymes in this complex are much more complex and regulate both M1 and M2 genes – the timing in the *in vivo* is relevant, hence seeing more time points, during the initial injury and during the repair phases is important.

We agree with the reviewer's comment. The role of H3K27me3 demethylases in the myeloid inflammatory-mediated cardiac repair after MI is critical, should not be overlooked and must be further elucidated as discussed in our manuscript. However, we believe that this constitute a complete independent new study from ours. Furthermore, even though the literature presenting H3K27me3 demethylases and more particularly, JMJD3 (*KDM6B*), as critical enzymes during M2 immunomodulatory macrophage phenotype differentiation is dated and controversial with the recent bibliography, it seems relevant in the context of cardiovascular diseases⁷. Indeed, publications from another group have revealed that the balance between the myeloid activity of JMJD3⁸ and EZH2⁹ regulates atherosclerosis development in general agreement with our findings.

To answer this comment, we decided to identify the H3K27me3 demethylase whose expression is the most affected by the polarization of M0 macrophages into M1 pro-inflammatory or M2 immunomodulatory macrophages for 2 days with LPS (50 ng/ml) or with a combination of both IL4 and IL10 (20 ng/ml) respectively *in vitro*. Surprisingly, we observed that UTX (*KDM6A*) expression was decreased during M0 to M1 polarization in both murine and human macrophages (Reviewer figure 1a and b respectively) while JMJD3 (*KDM6B*) remained unchanged (Reviewer figure 1c and d). The decrease in *KDM6A* expression was correlated with the increased expression of M1 pro-inflammatory specific macrophages markers (*IL1B* and *IL6*, Reviewer figure 1e-f and g-h respectively) and a decreased expression of M2 immunomodulatory specific macrophages markers (*PGF* and *MRC1*, Reviewer figure 1i-j and k-l respectively). The M2 immunomodulatory macrophage phenotype was not characterized by any significant modification of both *KDM6A* (Reviewer figure 1a and b) and *KDM6B* (Reviewer figure 1c and d) but confirmed by the down-regulation of M1 pro-inflammatory specific macrophages markers (*IL1B* and *IL6*, Reviewer figure 1e-f and g-h respectively) and the up-regulation of M2 immunomodulatory specific macrophages markers (*PGF* and *MRC1*, Reviewer figure 1i-j and k-l respectively). Altogether these data indicate that in our model of macrophage polarization, UTX (*KDM6A*) could be the more attractive EZH2 antagonistic partner to characterize as the H3K27me demethylase taking over its activity while it is translocated in the cytoplasm during macrophage polarization.

To answer the second part of this comment, and to better understand the importance of Ezh2 subcellular localization during macrophage differentiation post-MI, we performed new immunohistochemistry stainings for pro-inflammatory (*i.e.* Cd68⁺/Cd86⁺ in supplementary Figure S1,

Cd68⁺/MHCII⁺ in supplementary Figure S2, Cd68⁺/Il1b⁺ in supplementary Figure S3, Cd68⁺/Cd206⁻ in supplementary Figure S4, Cd68⁺/Trem2⁻ in supplementary Figure S5, Cd68⁺/Lyve1⁻ in supplementary Figure S6) and immunomodulatory (*i.e.* Cd68⁺/Cd86⁻ in supplementary Figure S1, Cd68⁺/MHCII⁻ in supplementary Figure S2, Cd68⁺/Il1b⁻ in supplementary Figure S3, Cd68⁺/Cd206⁺ in supplementary Figure S4, Cd68⁺/Trem2⁺ in supplementary Figure S5, Cd68⁺/Lyve1⁺ in supplementary Figure S6) macrophage phenotypes for multiple time points post-MI (Sham, an early 24h time point, a 72h intermediate time point and a 7 days late time point) corresponding to critical time points previously describes during post-MI inflammatory kinetic¹⁻³. In agreement with our data, Ezh2 is predominantly located in the nucleus of pro-inflammatory macrophages (Cd68⁺/Cd86⁺ supplementary Figure S1, Cd68⁺/MHCII⁺ supplementary Figure S2, Cd68⁺/Il1b⁺ supplementary Figure S3, Cd68⁺/Cd206⁻ supplementary Figure S4 and Cd68⁺/Lyve1⁻ supplementary Figure S6) at both early (24h) and intermediate (72h) time points post-MI. As expected, Ezh2 is translocated in the cytoplasm of immunomodulatory (both resident in sham mice and differentiated macrophages mostly from 72h to 7 days post-MI) macrophages (Cd68⁺/Cd86⁻ supplementary Figure S1, Cd68⁺/MHCII⁻ supplementary Figure S2, Cd68⁺/Il1b⁻ supplementary Figure S3, Cd68⁺/Cd206⁺ supplementary Figure S4 and Cd68⁺/Lyve1⁺ supplementary Figure S6).

Reviewer figure 1: Expression of UTX (*Kdm6a*), JMJD3 (*Kdm6b*), M1 (*Il1b* and *Il6*) and M2 (*Pgf* and *Mrc1*) macrophage polarization markers in the macrophage cell lineage.

Expression of *Kdm6a*, *Kdm6b*, *Il1b*, *Il6*, *Pgf* and *Mrc1* in mouse bone marrow-derived macrophages (a, c, e, g, i, k) and human peripheral blood-derived macrophages (b, d, f, h, j, l) non-polarized (M0) or polarized for 2 days either in pro-inflammatory macrophages (M1) with LPS (50 ng/ml) or immunomodulatory macrophages (M2) with a combination of both IL4 and IL10 (20 ng/ml) *in vitro* was measured by RT-qPCR. Data are presented as mean expression reported to M0 macrophages \pm SEM with *B2M* serving as internal control of 6 independent mice (n=6) and 3 to 4 independent human healthy donors (n=3 to 4) in duplicate. The p values are depicted as asterisks in the figures as follows: ***p <0.01; **p <0.01; *p <0.05 and ns: non-significant.

Reviewer #2 (Remarks to the Author):

This interesting manuscript examines epigenetic regulation of myeloid cells in the setting of infarct healing, specifically EZH2, responsible for H3K27 methylation. After MI and *in vitro*, EZH2 localizes in the cytoplasm of infarct macrophages, rendering it inactive. Pharmacological EZH2 inhibition reduced H3K27 methylation, affected macrophages gene expression and phenotypes, and improved post MI recovery. A number of *in vitro* studies were done with human monocytes, including from infarct patients, that were differentiated *in vitro*.

While interesting and potentially important, the study has some shortcomings. I can see 2 major limitations:

1. The data do not mechanistically show that EZH2 manipulation acts via macrophage phenotypes. While this is not unlikely, it isn't proven. This would be best accomplished with *in vivo* studies that delete EZH2 specifically in macrophages.

We thank the reviewer for raising this critical point in our study. We have considered this comment and established a mouse colony carrying a specific *Ezh2* deletion in the myeloid cell lineage (more particularly monocytes and mature macrophages) as advised by the reviewer. This was accomplished by the breeding of mice in which *Ezh2* was deleted specifically in the myeloid cell lineage under the control of the LysM (*Lyz2*) promoter named LysM-Cre^{+/-} *Ezh2* fl/fl and mice carrying only the LoxP sites flanking the *Ezh2* exons 16 and 17 to be deleted named LysM-Cre^{-/-} *Ezh2* fl/fl, serving as control mice for our *in vivo* experiments (Reviewer figure 2a). Genotyping of mice was determined by a PCR reaction for *Ezh2* loxP site downstream of exon 17 (Reviewer figure 2b), for *Lyz2* (LysM)-Cre knock-in/knock-out promoter allele (Reviewer figure 2c) and for *Ezh2* exons 16 and 17, indicating successful myeloid cell lineage deletion and recombination (Reviewer figure 2d). More information about this genetically-modified mouse model is displayed in the Reviewer methods-*Ezh2* myeloid deletion mouse model paragraph below.

We performed permanent left coronary artery ligation on both littermate control (LysM-Cre^{-/-} *Ezh2* fl/fl) mice and myeloid *Ezh2* deleted (LysM-Cre^{+/-} *Ezh2* fl/fl) mice to assess the implication of myeloid-selective *Ezh2* inhibition in the cardiac benefit observed with *Ezh2* inhibitor, GSK-343 treatment (Figure 6) post-MI. Surprisingly, and unexpectedly, we observed that *Ezh2*-specific deletion in myeloid cells significantly reduced mice survival after-MI in a dramatic manner (Reviewer figure 2e). We hypothesized that *Ezh2* myeloid deletion increases mice mortality after MI by preventing the occurrence of the cardiac acute inflammatory phase, well-known to be required for a proper cardiac healing after MI¹⁰ while the treatment with *Ezh2* inhibitor, GSK-343, is only accelerating its resolution. Thus, *Ezh2* genetic specific myeloid deletion might worsen the cardiac phenotype after-MI instead of improving it. To test this hypothesis, we characterized the cardiac inflammatory cell infiltration 3 days post- MI by flow cytometry. Interestingly we found that

the MI-induced granulocyte infiltration in LysM-Cre^{-/-} Ezh2 fl/fl mice is abolished by Ezh2-specific myeloid deletion in LysM-Cre^{+/-} Ezh2 fl/fl mice (Reviewer figure 2f). This phenotype might be explained by *Ezh2* deletion in LysM-expressing neutrophils, in addition to monocytes and macrophages,⁹. Indeed, *Ezh2* deletion in neutrophils may induce reduced adhesion and migration capacities¹¹. Thus, the LysM-Cre^{+/-} Ezh2 fl/fl mouse model of Ezh2 deletion in myeloid cell lineage is not sufficiently resolute to address selectively the role of Ezh2 in monocytes and macrophages. Further studies, using other myeloid cell lineage selective lineages (e.g. inducible Ezh2 myeloid deletion: Cx3cr1-CreERT2^{+/-} Ezh2fl/fl) are required. Interestingly, we similarly observed a decreased, albeit not completely abolished, granulocyte cardiac infiltration in our GSK-343 Ezh2 pharmacological inhibition treated mice, which might also explaining the difference observed between the two studies (Reviewer figure 2g).

As we have not been able to mechanistically show that EZH2 manipulation acts only via macrophage phenotypes, we have amended this critical observation in our main manuscript.

Editorial Note: This Figure was partly generated using Servier Medical Art, provided by Servier, licensed under a Creative Commons Attribution 3.0 unported license.

Reviewer figure 2: Ezh2 specific myeloid deletion impairs cardiac recovery after MI.

Breeding scheme used to produce myeloid selective *Ezh2* deleted mice (LysM-Cre^{+/-} *Ezh2* fl/fl) (a) and representative pictures of *Ezh2 loxP* (b), *Lyz2(LysM)-Cre* (c) and *myeloid deleted (del) Ezh2* (d) genotyping PCR results. Kaplan–Meier survival curves for littermate (LysM-Cre^{-/-} *Ezh2* fl/fl) or LysM-Cre^{+/-} *Ezh2* fl/fl mice after sham or permanent left coronary artery ligation-induced MI (e). The p values are depicted as asterisks (*) symbols for LysM-Cre^{+/-} *Ezh2* fl/fl MI versus LysM-Cre^{-/-} *Ezh2* fl/fl MI comparison or hashtag (#) symbols LysM-Cre^{+/-} *Ezh2* fl/fl MI versus LysM-Cre^{+/-} *Ezh2* fl/fl sham comparison in the figures as follows: * or # p <0.05; ns: non-significant. Cardiac granulocytes (Cd11c^{neg}Cd11b^{hi}Ly6c^{med}Cx3cr1^{neg}) population frequencies 3 days post-MI were evaluated by flow cytometry from either sham or MI LysM-Cre^{-/-} *Ezh2* fl/fl or LysM-Cre^{+/-} *Ezh2* fl/fl mice (f) and sham or MI mice treated daily either with vehicle (captisol 20%) or GSK-343 (g). Data are expressed as mean frequency ± SEM of live CD45⁺ cells from 3 to 4 independent biological replicates (n=3 to 4). The p values are depicted as asterisks in the figures as follows: *p <0.05 and ns: non-significant

2. The second major limitation is the overuse of the M1/M2 nomenclature and reliance on *in vitro* data. Admittedly, the M1/M2 nomenclature is valid for *in vitro* experiments, but does not really reproduce in the infarct. Studies have shown that certain M1/M2 genes are regulated in parallel, and that the macrophage subsets or phenotypes are more complex than what is seen on plastic, which lacks the typical microenvironment.

We thank the reviewer for this relevant comment. We have modified the M1/M2 macrophage nomenclature and now use pro-inflammatory versus immunomodulatory macrophages respectively, preceded by their expression marker characterization for our *in vivo* studies. Additionally, we now provide better characterization of these macrophage subsets, providing new immunohistochemistry staining to characterize the *in vivo* macrophage phenotypes (*i.e.* Cd68⁺/Cd86⁺, Cd68⁺/MHCII⁺, Cd68⁺/Il1b⁺ for pro-inflammatory macrophages and Cd68⁺/Trem2⁺, Cd68⁺/Lyve1⁺ for immunomodulatory macrophages) as shown in new Figure 1 panels c, d and g.

Altogether, we have investigated the subcellular localization of *Ezh2* in no less than 4 different subsets of pro-inflammatory macrophages (*i.e.* Cd68⁺/*i*Nos⁺, Cd68⁺/Cd86⁺, Cd68⁺/MHCII⁺, Cd68⁺/Il1b⁺) and 3 different subsets of immunomodulatory macrophages (*i.e.* Cd68⁺/Cd206⁺, Cd68⁺/Trem2⁺, Cd68⁺/Lyve1⁺) as well as their corresponding negative phenotypes (Figure 1 and supplementary Figures S1 to S6). We have also studied *Ezh2* subcellular localization upon heart myeloid cells kinetics post-MI within these macrophage phenotypes (supplementary Figures S1 to S6) from sham to 7 days post-MI with two intermediate time points 24h and 72h post-MI.

As expected from our previous observations, we confirm that *Ezh2* is predominantly located in the nuclei of both monocytes (Figure 1a) and pro-inflammatory cardiac macrophages at early time points post-MI Cd68⁺/*i*Nos⁺ in Figure 1b, Cd68⁺/Cd86⁺ in Figure 1c and supplementary Figure S1, Cd68⁺/MHCII⁺ in Figure 1d and supplementary Figure S2, Cd68⁺/Il1b⁺ in supplementary Figure S3,

Cd68⁺/Cd206⁻ Figure 1e and supplementary Figure S4 and Cd68⁺/Lyve1⁻ in supplementary Figure S6), while it is predominantly translocated in the cytoplasm of immunomodulatory cardiac macrophages at late time points after MI and in sham mice (Cd68⁺/iNos⁻ in Figure 1b, Cd68⁺/Cd86⁻ in Figure 1c and supplementary Figure S1, Cd68⁺/MHCII⁻ in Figure 1d and supplementary Figure S2, Cd68⁺/Il1b⁻ in supplementary Figure S3, Cd68⁺/Cd206⁺ Figure 1e, f and supplementary Figure S4 and Cd68⁺/Lyve1⁺ in Figure 1g and supplementary Figure S6) *in vivo*.

Minor comments:

1. Fig. 1: macrophages are often autofluorescent. Please provide controls to exclude this is the case.

We acknowledge the reviewer for pointing out this omission. We now provide negative autofluorescence controls for Cd68/Ezh2/Cd86 in the revised manuscript (Figure 1c and supplementary Figure S1a), Cd68/Ezh2/MHCII (Figure 1d and supplementary Figure S2a), Cd68/Ezh2/Cd206 (Figure 1e, f and supplementary Figure S4a), Cd68/Ezh2/Lyve1 (Figure 1g and supplementary Figure S6a), Cd68/Ezh2/Il1b (Supplementary Figure S3a) and Cd68/Ezh2/Trem2 (supplementary Figure S5a).

2. Fig. 3b and 4e: FDR values would be more appropriate than p values

In agreement with the reviewer's comment, we have modified all of figures depicting bio-informatic analysis to interchange p-value with more relevant and commonly accepted Benjamini p-value (Figure 3b, Figure 4e, supplementary Figure S10)

3. Fig. 6: please provide FACS plots to illustrate gating

We apologize for omitting to describe our flow cytometry gating strategy to analyse blood and heart samples. We have now included a new supplementary Figure S14 to better describe it for both mouse blood (a) and heart (b) samples.

4. Fig. S4: differences may become significant in larger cohorts (only 7 infarct patients). However, this is a minor point since these data are not novel.

We agree with the reviewer's comment about the tendency of an increased number of circulating monocytes in patients after MI. The number of patients (n=7) included in the cohort is indeed too low to obtain a potential significant difference as pointed out. However, we have modified the figure to show the heterogeneity of the data distribution, suggesting that blood circulating monocyte frequency seems to be similar between the three groups of patients.

5. Supplemental Tab 5: Please provide example histology images for the respective stains and groups

As suggested by the Reviewer, we provide pictures for each of the stainings in sham as well as 3- and 8-days post-MI following treatment with either vehicle or GSK-343 in the revised manuscript (supplementary Figure S15).

Reviewer methods

Ezh2 myeloid deletion mouse model

Lyz2-Cre^{+/-} (LysM-Cre^{+/-}, B6.129P2-Lyz2^{tm1^(cre)lfo}/J, The Jackson Laboratory Cat#004781) and *Ezh2^{fl/fl}* (B6;129S1-Ezh2^{tm2^{Sho}}/J, The Jackson Laboratory Cat#022616) murine breeders were generously given by Pr. Menno P. J. de Winther and Anette E Neele from the department of medical biochemistry, Amsterdam cardiovascular sciences, Amsterdam infection and immunity, Amsterdam UMC, University of Amsterdam, Amsterdam, Netherlands. Myeloid cells-specific Ezh2 exons 16 and 17 deficient mice (LysM-Cre +/- Ezh2 fl/fl) were obtained from a breeding with LysM-Cre +/- Ezh2 fl/fl and LysM-Cre -/- Ezh2 fl/fl (Reviewer figure 2a). The genotype of each mouse was verified using the following primers after mouse ear punch genomic DNA extraction. Insertion of LoxP sites in the *Ezh2* gene was assessed by *Ezh2 loxP* PCR using the following primers: *Ezh2^{fl/fl}* int17 forward primer: 5'-CATGTGCAGCTTTCTGTTCA-3' and *Ezh2^{fl/fl}* int17 reverse primer: 5'-CACAGCCTTTCTGCTCACTG-3' (WT 203 bp and flox ~ 300 bp amplicon, reviewer figure 2b). Knock-in insertion of Cre sequence within the promoter of the murine *Lyz2* gene was observed with the *Lyz2*(LysM)-Cre PCR using the following primers : *Lyz2*-WT forward primer the 5'-CTTGGGCTGCCAGAATTTCTC-3' and *Lyz2*-WT reverse primer: 5'-TTACAGTCGGCCAGGCTGAC-3' (346 bp amplicon) ; *Lyz2*-Cre forward primer: 5'-CTTGGGCTGCCAGAATTTCTC-3' (*Lyz2*-WT forward primer) and *Lyz2*-Cre reverse primer: 5'-CCCAGAAATGCCAGATTACG-3' (700 bp amplicon, reviewer figure c). Finally, myeloid cells-specific Ezh2 exons 16 and 17 deletion (*Ezh2*LysM del ex16-17) was visualized using the previously used flox primers sequence: *Ezh2^{fl/fl}* int15 forward primer: 5'-CCCATGTTTAAGGGCATAGTG-3' and *Ezh2^{fl/fl}* int17 reverse primer: 5'-CACAGCCTTTCTGCTCACTG-3' (WT 2117 bp and *Ezh2*del ex16-17 ~ 700 bp amplicon, reviewer figure 2d). The reviewer figure (2b-d) a provides examples of each of these genotypes.

RT-qPCR additional primers

Gene and location	Primer sense	Primer sequence (5' > 3')	Amplicon size
KDM6A/ Kdm6a	Forward	AGCGCAAAGGAGCCGTGGAAAA	97 bp
	Reverse	GTCGTTCAACCATTAGGACCTGC	
KDM6B	Forward	GACCCTCGAAATCCCATCACAG	122 bp
	Reverse	GTGCGAACTTCCACGGTGTGTT	
Kdm6b	Forward	AGACCTCACCATCAGCCACTGT	116 bp
	Reverse	TCTTGGGTTTCACAGACTGGGC	

Bibliography

1. Rizzo G, Gropper J, Piollet M, et al. Dynamics of monocyte-derived macrophage diversity in experimental myocardial infarction. *Cardiovasc Res*. Aug 11 2022;doi:10.1093/cvr/cvac113
2. Mouton AJ, DeLeon-Pennell KY, Rivera Gonzalez OJ, et al. Mapping macrophage polarization over the myocardial infarction time continuum. *Basic Res Cardiol*. 06 2018;113(4):26. doi:10.1007/s00395-018-0686-x
3. Forte E, Furtado MB, Rosenthal N. The interstitium in cardiac repair: role of the immune-stromal cell interplay. *Nat Rev Cardiol*. Oct 2018;15(10):601-616. doi:10.1038/s41569-018-0077-x
4. Noels H, Weber C, Koenen RR. Chemokines as Therapeutic Targets in Cardiovascular Disease. *Arterioscler Thromb Vasc Biol*. Apr 2019;39(4):583-592. doi:10.1161/ATVBAHA.118.312037
5. Dusi V, Ghidoni A, Ravera A, De Ferrari GM, Calvillo L. Corrigendum to "Chemokines and Heart Disease: A Network Connecting Cardiovascular Biology to Immune and Autonomic Nervous Systems". *Mediators Inflamm*. 2018;2018:4128049. doi:10.1155/2018/4128049
6. Nossent AY, Bastiaansen AJ, Peters EA, et al. CCR7-CCL19/CCL21 Axis is Essential for Effective Arteriogenesis in a Murine Model of Hindlimb Ischemia. *J Am Heart Assoc*. Mar 2017;6(3)doi:10.1161/JAHA.116.005281
7. Yuan JL, Yin CY, Li YZ, Song S, Fang GJ, Wang QS. EZH2 as an Epigenetic Regulator of Cardiovascular Development and Diseases. *J Cardiovasc Pharmacol*. Aug 01 2021;78(2):192-201. doi:10.1097/FJC.0000000000001062
8. Neele AE, Gijbels MJJ, van der Velden S, et al. Myeloid Kdm6b deficiency results in advanced atherosclerosis. *Atherosclerosis*. Aug 2018;275:156-165. doi:10.1016/j.atherosclerosis.2018.05.052
9. Neele AE, Chen HJ, Gijbels MJJ, et al. Myeloid Ezh2 Deficiency Limits Atherosclerosis Development. *Front Immunol*. 2020;11:594603. doi:10.3389/fimmu.2020.594603
10. van Amerongen MJ, Harmsen MC, van Rooijen N, Petersen AH, van Luyn MJ. Macrophage depletion impairs wound healing and increases left ventricular remodeling after myocardial injury in mice. *Am J Pathol*. Mar 2007;170(3):818-29. doi:10.2353/ajpath.2007.060547
11. Gunawan M, Venkatesan N, Loh JT, et al. The methyltransferase Ezh2 controls cell adhesion and migration through direct methylation of the extranuclear regulatory protein talin. *Nat Immunol*. May 2015;16(5):505-16. doi:10.1038/ni.3125

Although we partially answered reviewer#1 question 5 and reviewer#2 major comment 1 ; we deeply believe that our manuscript has strongly been improved in the revised version. All authors approved the revised manuscript for re-submission, and declare they have no conflict of interest. RNA-Seq and CHIP-seq data have been deposited into the public GEO database (GSE165543 and GSE226811 respectively) with protected access for reviewers using link and provided token.

RNA-seq data:

Link: <https://www.ncbi.nlm.nih.gov/geo/query/acc.cgi?acc=GSE165543>

Token: mvmvwsogdhszlez

CHIP-seq data:

Link: <https://www.ncbi.nlm.nih.gov/geo/query/acc.cgi?acc=GSE226811>

Token: szmfueoirhmvtd

REVIEWERS' COMMENTS

Reviewer #1 (Remarks to the Author):

They did an excellent job of responding to my prior comments and added new experiments and revised many of their cell markers in accordance with standards.

Reviewer #2 (Remarks to the Author):

The revision work is satisfactory.